# Allosteric modulation of *Plasmodium falciparum* Isoleucyl tRNA synthetase by South African natural compounds

Curtis Chepsiror [ID], Wayde Veldman, Fisayo Olotu, Özlem Tastan Bishop [ID]*

Research Unit in Bioinformatics (RUBi), Department of Biochemistry, Microbiology and Bioinformatics, Rhodes University, Makhanda, South Africa

* o.tastanbishop@ru.ac.za

## Abstract

Targeting *Plasmodium falciparum* (Pf) aminoacyl tRNA synthetases is a viable strategy to overcome malaria parasite multi-drug resistance. Here, we focused on *Pf* Isoleucyl tRNA synthetase (PfIleRS) to identify potential allosteric inhibitors from 1019 South African Natural Compounds (SANC). Eleven potential hits, which passed ADMET and PAINS, were selected based on their docking binding affinity which was higher for PfIleRS than for human IleRS. Molecular dynamics simulations revealed that the compounds, particularly SANC456, commonly induced considerable changes in the global conformation and dynamics of PfIleRS, suggesting potential allosteric modulatory effects. Importantly, all 11 SANC hits reduced the binding affinity of the nucleotide AMP molecule by at least 25%. Some SANC ligand-bound systems (SANC456, SANC1095, and SANC1104) significantly increased the distance between the AMP and Ile ligands. Possible explanations for these changes were explored using three dynamic residue network centrality metrics. *Betweenness centrality* identified a major allosteric pathway in holo PfIleRS spanning the entire protein length. In contrast, SANC382, SANC456, SANC522, SANC806 and SANC1095 ligand-bound systems exhibited delta *BC* pathways (SANC-protein minus holo-protein), induced by the ligands, extending from their respective pockets into the active site. Additionally, *eigenvector centrality* revealed two important residue clusters either side of the holo active site which became altered in the ligand-bound systems, indicating possible allosteric activity. Lastly, many SANC systems showed decreased *closeness centrality* of zinc finger and active site residues, including the HYGH and KMSKR motifs. We believe that the compounds identified in this study as potential allosteric inhibitors have strong translational potential and warrant further investigation through *in vitro* and *in vivo* experiments. Overall, they hold promise as starting points for the development of new and effective antimalarial therapies, particularly against multidrug-resistant Plasmodium parasites.

**Data availability statement:** All relevant data are within the paper and its Supporting information files. Compounds are available at https://sancdb.rubi.ru.ac.za/. Scripts (Docking, MD, MM-GBSA, Delta analysis for RMSF, DRN and MM-GBSA with PyMOL visualization) and MD trajectories, PyMOL sessions are available at: https://doi.org/10.6084/m9.figshare.28644392.v1.

**Funding:** The author(s) received the following funding for this work: C.C. was supported as an MSc student by the Queen Elizabeth Commonwealth Scholarships (QECS) (Award number: 14561). W.V. and F.O. were funded as postdoctoral fellows by the South African National Research Foundation under grant numbers 138566 and PSTD2204214166, respectively. W.V. is currently a recipient of the Rhodes University Postdoctoral Fellowship.

**Competing interests:** The authors have declared that no competing interests exist.

## 1. Introduction

Malaria is an infectious disease of serious public health concern in many parts of the world. We have a limited number of drugs in use, and these drugs often present challenges, with multi-drug resistance being the most significant one. Artemisinin-based combination therapies (ACTs) are currently the first-line antimalarial drugs [1], and have substantially decreased malaria associated mortality and morbidity from 0.5 million deaths in 2014 to 0.247 million as reported in 2023 [2,3]. However, malarial parasite *Plasmodium falciparum* (*Pf*) has been gradually gaining resistance to ACTs, first detected in Asia and more recently in Africa [4]. The second challenge is adverse drug reactions. While ACTs generally have no dose-related adverse effects in humans, other commonly used antimalarial drugs like mefloquine can cause serious neuropsychiatric toxicity [5].

In this article, we identified potential hit compounds against cytoplasmic isoleucyl tRNA synthetase (IleRS; EC 6.1.1.5) of *Pf* while considering drug resistance and drug adverse reaction issues. One potential strategy to reduce the development of drug resistance is to target multiple stages of the parasite's life cycle. As discussed in our recent articles, aminoacyl tRNA synthetases (aaRSs) play central roles in the synthesis of the Plasmodium proteome at every life-stage of the parasite [6,7], and IleRS is no exception. Additionally, cytoplasmic PfIleRS has already been validated as a multi-stage antimalarial target [8]. Furthermore, we aim to reduce potential off-target toxicity by focusing on the identification of allosteric modulators [9,10]. Allosteric sites are less conserved compared to orthosteric (functional) sites, making allosteric modulators highly specific and potentially less toxic to the host [11]. Allosteric drugs can also remain effective even in the presence of native substrates, reducing the likelihood of resistance development by parasites through increased substrate concentrations. When allosteric drugs lack agonistic effects and are only active in the presence of the substrate, they preserve the spatial and temporal activity of the endogenous substrate [11–14]. Additionally, studies have demonstrated that allosteric modulators can be easily modified to enhance their activity, unlike orthosteric drugs [11,12]. Notably, no allosteric drug has yet been developed as an antimalarial.

We also screened the protein against natural compounds, known to provide evolutionary optimised large scaffold diversity, hence opening new avenues for identifying new potential hit compounds [15]. Previously, mupirocin, a natural product from *Pseudomonas fluorescens,* had been used to inhibit bacterial apicoplast IleRS [16,17], and has been further shown to inhibit blood stage PfIleRS [18]. Fused pyrimidines have also been of great importance due to their wide spectrum of activity in drug discovery. Thienopyrimidines which are a subclass of pyrimidines have shown multistage activity against PfIleRS [8,19].

Cytoplasmic PfIleRS is a class Ia monomeric aaRS, and overall structure consists of Rossmann-fold catalytic domain (residues 93–254, 707–824), connective polypeptide (CP) core (residues 255–279, 579–599), CP1 editing domain (residues 280–578), CP2 (residues 600–639), CP3 (residues 640–706), anticodon-binding domain (residues 825–1014), and C-terminal junction domain (residues 1015–1181) (Fig 1A and 1B) [6,20]. We identified the start and end residues of these domains, which

differ slightly from those reported by Istvan [8]. This was achieved by extrapolating the crystal structure information [20] to our calculated multiple sequence alignment (Fig 1C). PfIleRS also has zinc finger (ZnF) motifs formed due to hydrogen bonding between residues S265, C266, K267, and C268 (ZnF1), and C579, W580, R581, and S582 (ZnF2) which is crucial for interdomain communication and structural conformation [8,21,22]. The N and C-terminal Rossmann folds are the catalytic domain (active pocket), separated by Connective Peptide 1, 2, and 3. The editing domain is used in proofreading to ensure the fidelity of aminoacylation reactions by hydrolysing mis-activated and mis-charged amino acids and tRNAs, respectively [23]. The anticodon binding domain is the C-terminal tRNA stem-acceptor loop, which anchors the cognate tRNA to the enzyme [24]. PfIleRS also contains KMSKR (residues 783–787) and HYGH (residues 138–141) conserved sequences which are essential in influencing the conformation of the active pocket [25,26]. Furthermore, PfIleRS has several potential druggable pockets some of which have been previously reported [24,27], including the ATP binding pocket, anticodon binding pocket, amino acid binding pocket, editing domain, and also the potential allosteric pockets which were predicted in this study.

Here, 1019 natural compounds from the South African Natural Compounds Database (SANCDB; https://sancdb.rubi.ru.ac.za/) [30,31] were screened against plasmodium and human IleRS proteins to identify potential allosteric modulators. Potential binding pockets were further validated using prediction algorithms, and druggability attributes of all predicted pockets were assessed before considering them as potential allosteric pockets. Blind compound docking was employed to predict potential hits that can bind more selectively to these pockets in plasmodium over

**Fig 1. PfIleRS model from *Saccharomyces cerevisiae* and *Candida albicans* templates.** (A) The colour bar showing the residue ranges defining each domain. (B) Homology model of PfIleRS showing cartoon representation of the functional domains: N-terminal Rossmann fold (NT-RF; blue), Zinc finger 1 hinge (ZnF1; yellow), Connective Peptide I (Editing domain; red), Zinc finger 2 hinge (ZnF2; yellow), Connective Peptide 2 and 3 (olive and orange), C-terminal Rossmann fold (CT-RF), Anticodon Binding Domain (ABD; purple) and Junctional domain (cyan). The HYGH and KMSKR signature motifs are shown in green and magenta spheres, respectively. (C) Multiple sequence alignment of PfIleRS target with 7D5C and 6LDK templates showing secondary structure predictions (number of alpha helices and beta sheets across the PfIleRS reference structure) with HYGH, ZnF1, ZnF2 and KMSKR conserved motifs in black blocks; the asterisks indicate the start and end of the modelling sequence [20,28,29].

the human homolog. Based on predicted binding affinities, eleven compounds that had preferential binding to PfIleRS over human IleRS were selected for further analysis. Molecular dynamics (MD) simulations showed that all the identified hit compounds were stable within the protein throughout the simulations. Four SANCDB potential hits (SANC140, SANC456, SANC968, and SANC1095) that were bound to predicted allosteric pocket 1, located at the Rossman fold region, introduced considerable conformational changes in the PfIleRS global structure, an indication that these compounds have allosteric effects in the protein. Further, free binding energy calculations revealed that all potential inhibitors reduced the binding affinity of AMP nucleotide. Dynamic residue network (DRN) analysis revealed important residues forming communication pathways or communities in the functional domains of the protein, with SANC ligand binding changing these pathways or communities, which is an indication of allosteric modulation. For instance, SANC806 ligand binding increased the *betweenness centrality* (*BC*) of residues that form a path from its potential allosteric pocket located in the editing domain, through the Rossmann fold and anticodon binding domains, to the junctional domain located at the opposite end of the protein. On the other hand, delta *eigenvector centrality (EC)* revealed important groups of residues around the active site with increased/decreased *EC* values, induced by some SANC ligands. Overall, we established a computational approach applicable to other allosteric inhibitor identification projects, and it combines allosteric pocket identification, molecular docking, MD simulations, binding free energy calculations and DRN analysis to identify potential allosteric modulators and to demonstrate both global effects of allosteric modulators and changes at the residue level. We further propose that the allosteric hit compounds reported in this study can provide a viable starting point for the development of novel multi-stage antimalarial therapies.

## 2. Materials and methods

### 2.1. *Homology modelling*

Protein sequence of cytoplasmic PfIleRS (accession number: Q8IDZ9; 3D7 strain) and human homolog *Homo sapiens* isoleucyl tRNA synthetase (HsIleRS) (accession number: P41252) were retrieved from the UniProt database [32]. PRIMO [33] and HHpred [34] were used to identify good quality templates. For PfIleRS model, *Saccharomyces cerevisiae* (PDB ID: 7D5C) [20] with sequence similarity, coverage, and identity percentages of 76.7%, 88.9%, and 51.5%, and *Candida albicans* (PDB ID: 6LDK) [28] with 72.2%, 66.3%, and 39.1% for the same metrics, with respect to the PfIleRS sequence, were selected. The AlphaFold structure (ID: AF-Q8IDZ9) was also selected as a template to model the C-terminal of PfIleRS which was partially missing in the crystal structures. The quality of templates was further validated via Protein Data Bank (PDB) validation data and validation charts. 6LDK and 7D5C had resolutions of 1.9 Å and 2.9 Å with Rfree of 0.20, and 0.24 respectively (S1 Fig).

   *Candida albicans* (PDB ID: 6LDK) template with 66.7%, 67.3%, and 50.3% for sequence similarity, coverage, and identity respectively, with respect to HsIleRS, was utilised in modelling HsIleRS structure. The F300→A387 loop was chopped from PfIleRS model structure and residues in 3D structure renumbered to reflect the actual UniProt sequence numbering. Multiple sequence alignment (MSA) of each protein was carried out using Profile Multiple Alignment with Local Structures and 3D Constraints (PROMALS3D) [35] with modelling pir files generated using Jalview [36]. Modelling was done using the MODELLER v10.2 program [37] Python script using Automodel with slow refinement options. Plasmodium and human proteins were modelled with catalytic substrates (Ile and AMP) in the active pocket. 100 models were generated per protein and three models with minimum z-DOPE (Discrete Optimised Protein Energy) score were selected for validation. Further model validations were performed using Qualitative Model Energy Analysis (QMEAN) [38], Protein Structure Analysis (PROSA) [39] and Verify3D [40] web servers. One best model for PfIleRS and HsIleRS that passed these assessment metrics were selected for allosteric pockets prediction and molecular docking.

## 2.2. *Potential allosteric pocket prediction and conservation analysis*

The following tools were used for potential allosteric pocket prediction: Schrodinger SiteMap [41], DogSiteScorer [42], FTMap [43] and Passer [44]. SiteMap was the primary tool due to its detailed analysis of indicators such as druggability, SiteScore, pocket size, pocket volume, hydrophilicity, and hydrophobicity. SiteMap input settings were adjusted to report at least 15 grid-points per predicted pocket and identify a maximum of ten potential pockets. Our selection criteria was based on pockets identified across at least three prediction algorithms. Identified pockets included both the active site pocket of each protein and potential allosteric pockets. An in-house Python script was utilised to select residues within 5 Å of the potential allosteric pocket centroid to enhance prediction accuracy.

These pockets were further analysed based on residue conservation. Using the PROMALS3D web server [35] sequences of HsIleRS, PfIleRS, *Plasmodium malariae* IleRS (PmIleRS, UniProt ID: A0A1D3PAW4), *Plasmodium knowlesi* IleRS (PkIleRS, UniProt ID: A0A1Y3DUT2), *Mus musculus* IleRS (MmIleRS, UniProt ID: Q8BU30) were aligned to identify residues defining active site pocket and potential allosteric pockets. Visualisation was done using Jalview [36]. ConSurf server [45] was utilised with default options to generate conservation scores for the predicted pockets based on template structures. ConSurf calculates these scores using the RateSite algorithm [46] based on sequence identity of the compared protein structures. Scores were normalised using the Min-Max scaling method [47].

## 2.3. *Molecular docking*

1019 SANCDB compounds (https://sancdb.rubi.ru.ac.za/) were retrieved in PDBQT format, minimised and ready-to-dock; for structure-based high throughput virtual screening against PfIleRS and HsIleRS proteins [30,31].

**2.3.1. Protein preparation.** PDB2PQR [48] was used to protonate PfIleRS and HsIleRS proteins at a pH of 7.5 [8]. AMP was assigned a charge of -2 using Biovia Discovery Studio [49] (-O, =O, and -O dihedral atoms). Obabel function in MGLTools was used for receptor files conversion to formats compatible with AutoDock Vina [50,51].

**2.3.2. Blind docking and validation.** Docking parameters were optimised using AutoDock Vina with grid box sizes defined based on receptor dimensions in Chimera [52] (S1 Table). Redocking of AMP to the orthosteric pockets of PfIleRS and HsIleRS was used to determine optimum exhaustiveness for docking experiments. In all docking experiments, ligands were allowed to be flexible to enable screening of the entire protein structure (blind docking). For analysis of redocking validation, RMSD for the best redocked AMP and modelled AMP were determined for PfIleRS and HsIleRS using PyMOL molecular graphics tool [53]. LigPlot+ was used to analyse common residues interacting between the modelled and redocked AMPs and the receptors [54].

The accuracy of AutoDock Vina protocol was further tested using eleven known active inhibitors against PfIleRS, assembled from literature [2,8]. A total of 550 decoys for these compounds were generated from DUD-E (a directory of useful decoys) [55]. The decoys and bioactive compounds were docked to PfIleRS using AutoDock Vina followed by enrichment analysis using the Screening Explorer web server [56]. Receiver Operating Characteristic (ROC) and BEDROC curves were used to assess the level of sensitivity and accuracy of AutoDock Vina to identify active inhibitors.

The first round of docking was blind docking of 1019 SANCDB compounds to PfIleRS, and only ligands that docked to the predicted allosteric pockets were extracted using an in-house Python script, and the ones binding to the active pocket were not selected. The selected compounds, then, docked to HsIleRS. Those that were binding to the orthosteric pocket of HsIleRS protein were also discarded. Selective ligands to PfIleRS were determined based on initial hits criteria (those that only docked to PfIleRS) and preferential binding scores. Further, Absorption, Distribution, Metabolism, Excretion, and Toxicity (ADMET) properties were assessed using the SwissADME web server (http://www.swissadme.ch) for all SANCDB potential hits. Discovery Studio Visualiser was used to analyse 2D protein-ligand interactions of all selected ligands [49].

 

## 2.4. Molecular dynamics (MD) simulations

Selective potential hits from molecular docking of SANCDB compounds formed the initial structures for MD simulations. During initial optimisation to establish proper parameters for setting up the MD simulations of PfIleRS, we noted that the C-terminal domain was highly flexible, making it difficult to achieve convergence in the three holo systems. This dynamic nature of the extended PfIleRS C-terminal complicated the analysis of the global structure making it difficult to distinguish between meaningful structural changes from random events. Therefore, we truncated the flexible C-terminal of PfIleRS and consistency in the holo systems was achieved. Three replicates of PfIleRS and HsIleRS holo systems were performed to ensure consistency and to compare with PfIleRS-SANC and HsIleRS-SANC complexes. All-atom MD simulations were carried out using AMBER18 software suite [57]. Antechamber and GAFF [58,59] tools were used for ligand parameterisation. These tools generate files readable by the subsequent *tleap* program [60]. Antechamber and *parmchk* modules were used to generate ligand parameter files and respective net charges were assigned using AM1-BCC2 method on FF14SB force field [60]. The *pdb4amber* tool prepared PfIleRS and HsIleRS receptors, where all protonated histidine residues were renamed (HIS→HIP) for topology and coordinate file generation. *Tleap* was then used to generate topology and co-ordinate files for complex, receptors, and the ligands. TIP3P water model and solvate box size 12 Å were used with $Cl^-$ and $Na^+$ added to neutralise all the systems.

Partial minimisation for 5000 steps of steepest descent with a constant periodic boundary volume within a cutoff of 10 Å was carried out. A force constant of 10 $Kcal^{-1}$ $angstrom^{-2}$ was applied to all receptor residues. This was followed by full minimisation for 2500 steps without positional restraints, which allowed the whole system to minimise. The system was then allowed to heat up from 0K to 310K in reference to normal body temperature. To ensure that this happens without any wild fluctuations, weak positional restraints were applied. Initial random velocities were generated using Boltzmann distribution, and Langevin dynamics controlled the temperature with a collision frequency of 1.0 $ps^{-1}$. The system was set to harvest output files after every 100 steps with integration time step of 0.0005 ps. This ensured heating took place steadily without any drastic changes for 50,000 steps. Constant pressure was used to carry out equilibration to relax the density of water for 50,000 steps. Isotropic scaling was used to maintain pressure and temperature at 310K ($Temp_o = 310$, $Temp_i = 310$). Thus, MD simulations were initialised from where equilibration ended. Constant pressure with an average of 1 atm was applied in MD simulations. Non-bonded cutoff of 10 Å was used with no positional restraints. The SHAKE algorithm was used to constrain all bonds involving hydrogen. Temperature was maintained at 310K ($Temp_0 = 310$, $Temp_i = 310$) throughout the simulation. Each system underwent a 100 ns simulation, with output, trajectory, and restart files written every 5000 steps. All MD simulations were conducted at the Centre for High Performance Computing (CHPC) Cape Town.

## 2.5. Post MD analysis

AMBER18 integrated *Process_mdout* script was used to generate values for temperature, pressure, kinetic energy, total energy, and potential energy to monitor if the systems were progressing well. The AMBER18-integrated *cpptraj* tool [61] and *cpptraj-trajin* command were used to combine all trajectory files generated. AMBER *cpptraj Autoimage* and *strip* commands were used to align topology and trajectory files by stripping out water and ions before generating all metric values: root mean square deviation (RMSD), and root mean square fluctuation (RMSF). C-alpha carbon using AMBER *cpptraj-rms* CA was used to trace the position of the structure in reference to its initial position (RMSD). Average atomic fluctuations in reference to all other atoms were determined using AMBER cpptraj-*radgyr* command (RMSF). AMBER *cpptraj cluster* was used for calculating ensemble conformations of holo and protein-ligand complexes. AMBER *cpptraj com distance* was used to calculate the centre of mass (CoM) distance between Ile and AMP ligands. Data metrics calculated were analysed using Python libraries (Matplotlib, Pandas, Numpy, Seaborn, and MDAnalysis) to generate RMSD, RMSF, and CoM distance plots.

## 2.6. Binding free energy calculations

Molecular Mechanics Generalised-Born Surface Area (MM GBSA) was used to gain further insight into the stability of each receptor-ligand complex using AMBER18 *MM/PB(GB)SA* [62]. The Amber FF14SB force field was utilised with the last 2000 frames of the equilibrated trajectory. The following equations were applied in the calculation of respective delta energies:

$$\Delta G^{bind} = G^{complex} - \left( G^{receptor} + G^{inhibitor} \right) \tag{1}$$

$$\Delta G^{bind} = \Delta G^{gas} + \Delta G^{solv} - T\Delta S = \Delta H - T\Delta S \tag{2}$$

$$\Delta G^{gas} = \Delta E^{int} + \Delta E^{ele} + \Delta E^{vdW} \tag{3}$$

$$\Delta G^{solv} = \Delta E^{gb} - \Delta E^{surf} \tag{4}$$

$$\Delta E^{surf} = \gamma SASA + \beta \tag{5}$$

where, internal ($\Delta E^{int}$), electrostatic ($\Delta E^{ele}$), and van der Waals ($\Delta E^{vdW}$) constituted the gas phase energy ($\Delta G^{gas}$) while the polar solvation ($\Delta E^{gb}$), and non-polar contribution to solvation ($\Delta E^{surf}$) terms define solvation free energy ($\Delta G^{solv}$). $\Delta E^{gb}$ was estimated using the linear relationship between the surface tension proportionality constant ($\gamma = 0.0072\,mol^{-1}Å^{-2}$) in MM GBSA while $\Delta E^{surf}$ term was estimated using solvent accessible surface area (SASA), and $\beta$ constant. In addition, we calculated $\Delta G^{bind}$ for the nucleotide AMP molecule relative to the binding of the SANCDB hit compounds. Finally, per-residue energy decomposition was calculated using the *-idecomp* function to identify critical residues contributing to the estimated $\Delta G^{bind}$.

## 2.7. Dynamic residue network (DRN) analysis

DRN calculations were performed using MDM-TASK and MD-TASK scripts (https://github.com/RUBi-ZA/MD-TASK) [63,64] where DRN metrics are computed with a step size of 10 (every 10th MD frame of the trajectory), and the residue centrality values are averaged. The DRN metrics used were *averaged betweenness centrality* (*BC*), *averaged eigenvector centrality* (*EC*), and *averaged closeness centrality* (*CC*) calculated using the default cut-off (edge distance connecting two nodes) of 6.7 Å. In the rest of the article, *averaged BC, averaged CC*, and *averaged EC* will be referred to as *BC*, *CC*, and *EC*, respectively.

## 2.8. In-house delta Python scripts

During comparisons, delta calculations limit noise and clearly indicate major differences [65–67]. The holo enzyme was compared to all systems using delta in-house Python scripts. The steps are as follows:

**Step 1)** Per-residue RMSF, AMP ligand binding free energy, and DRN values were averaged over the holo replicates to obtain holo average values:

(PfIleRS$_i$-HoloAvr = (PfIleRS$_{i\_holo\_system1}$ + PfIleRS$_{i\_holo\_system2}$ + PfIleRS$_{i\_holo\_system3}$)/3; *i* = residue number, HoloAvr = Holo averaged)

**Step 2)** For each system, per-residue delta values were calculated by subtracting the holo average value from the corresponding residue of each system value:

(Delta$_{ij}$ = PfIleRS$_i$-SANCj - PfIleRS$_i$-HoloAvr; $i$ = residue number, $j$ = compound ID, SANC = compound, HoloAvr = Holo averaged)

**Step 3)** In each system, the residues with delta values in the highest and lowest 3% were identified (54 residues (highest and lowest 27 residues) per system).

**Step 4)** The 3% residues with the highest and lowest delta values per system (54 residues) were plotted in heatmaps with Seaborn [68] and mapped to the 3D enzyme structure using PyMOL [53]. The highest and lowest 3% delta residues (54 residues) are shown with an "**x**" in heatmaps and sticks or spheres on the structures. Residues in the heatmaps not shown with an "**x**" are not in the top/bottom 3% in their respective system; these are included for comparison purposes. The blue colour indicates a decrease in the specific metric value in the protein-ligand complex with respect to the holo-averaged protein for the specific residue, while the red colour indicates an increase. The colour intensities of the residues in the heatmaps and structures match for easy visual analysis. Unlike the RMSF and DRN delta calculations which used the top/bottom 3% residues per system, the AMP ligand binding free energy per-residue decomposition used the top/bottom 3 residues per system (not percent).

## 3. Results and discussion

### 3.1. *Potential allosteric pockets in Plasmodium falciparum and human IleRS proteins are identified*

Building on our previous work, this study expands the approach by integrating DRN analysis to identify the effect of novel potential allosteric modulators at atomic level [7]. Additionally, we further refined our calculations on both homology modelling and potential allosteric pocket detection. For the PfIleRS modelling, we included the C-terminal domain using the AlphaFold structure as an additional template. The best models had the z-DOPE scores of -1.29 and -1.30 for the *Pf* and human IleRS respectively indicating models with good-quality [69]. Further validation data is presented in S1 Fig.

For the allosteric pocket identification, in our previous study [7] default program parameters were used; here we optimised the input parameters, thus we report additional potential allosteric pockets. Furthermore, in addition to employing multiple prediction algorithms to detect potential pockets, we utilised residue conservation analysis to identify potent sites suitable for ligand binding (S2 Fig). Pocket residues are identified by using an in-house script and predicted pockets were used to guide the compound's selection process to identify potential hits. Information on the residue composition of each identified pocket is provided in S2 and S3 Tables, obtained by defining residues within 5 Å of the pocket centroid based on consensus from multiple prediction programs.

Aside from the active sites, seven pockets were identified in PfIleRS and only five in HsIleRS, unevenly distributed across three functional domains: the Rossmann fold, the editing domain, and the anticodon binding domain (Fig 2A and B). No targetable pocket was identified in the C-terminal domain. Pockets 1 and 2 are located in the Rossmann fold of the catalytic domain in both PfIleRS and HsIleRS proteins. Pockets 3 and 6 are located in the anticodon binding domain of PfIleRS, whereas HsIleRS has only pocket 3. Pockets 4, 5, and 7 are present in the editing domain of PfIleRS, while HsIleRS contains pockets 5 and 7. Pocket 8 as identified by all the prediction algorithms used correlates with the active site in both proteins (S2 and S3 Tables).

The features of each identified pocket were also assessed using SiteMap, including SiteScore, DScore, pocket size, pocket volume, hydrophilicity, and hydrophobicity, which together determine the suitability of the pocket for binding the compounds [70,71]. For instance, hydrophobic pockets favour the binding of drug-like molecules which usually contain hydrophobic moieties in their scaffolds and can displace ordered water molecules in such regions to form stable protein-ligand interactions [72]. In contrast, hydrophilic pockets are less suitable for ligand binding, as they may require charged functional groups that impede passive transport [73]. Further, pockets that are small (pocket size and volume) and less exposed may be unfavourable for ligand binding and therefore poorly druggable [41,74]. SiteScore and druggability score

(DScore) values were estimated based on these physicochemical properties and used to deduce the druggability potentials of the predicted allosteric pockets. As previously established, SiteScore and DScore values of 0.8 or above depict a pocket that has high ligand binding and druggability potentials [71,75].

Our results showed that all the predicted pockets (including active site pockets) had SiteScore and DScore values above 0.8 and pocket size greater than 100 Å (Fig 2C and D), thereby indicating their potential to favourably bind compounds which can be further exploited in any future studies. Among these pockets is one located in the Rossman fold which correlated with the region bounded by rotigotine, indecainide, carbamazepine, glycopyrronium, as well as the editing domain reportedly targeted by mitomycin [7]. Also, hydrophobicity scores were comparatively higher in the predicted allosteric sites than in the catalytic site which could indicate the presence of more hydrophobic residues in those regions [76].

Multiple sequence alignment of PfIleRS and HsIleRS further revealed differences in the amino acid compositions of these pockets (S2 Fig). Conserved allosteric residues in PfIleRS and HsIleRS exhibited structural and sequence-based quantitative differences suggesting that the predicted pockets possess unique features in Pf and human IleRS (S4 Table). Predicted pockets in PfIleRS and HsIleRS showed variations in physico-chemical properties in the conserved residues in

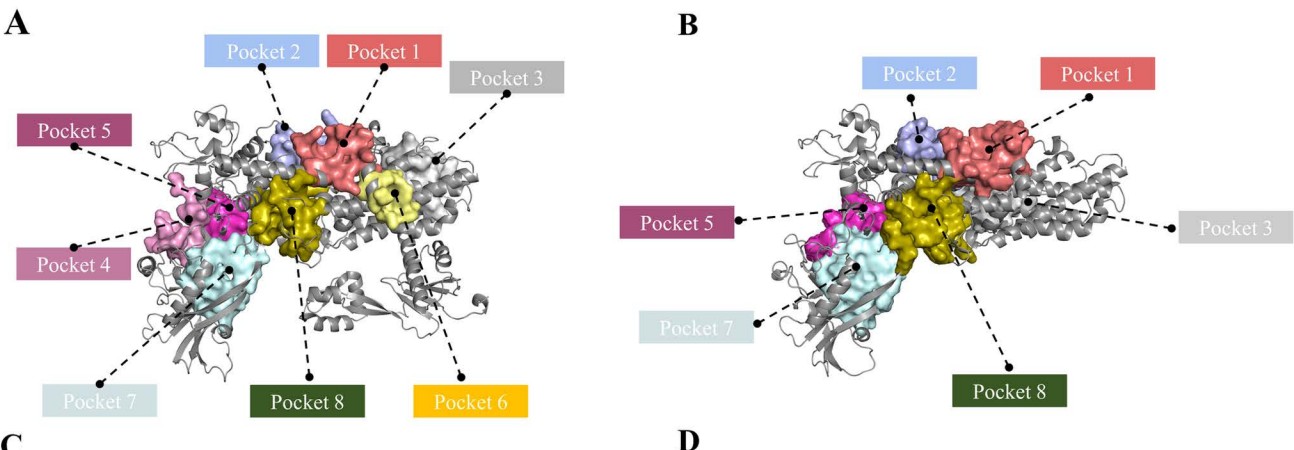

**C**

| Pocket | SiteScore | Size (Å) | DScore | Volume (A³) | Phobic (LogP) | Philic (LogP) |
|---|---|---|---|---|---|---|
| 1 | 1.06 | 153 | 0.95 | 257.59 | 1.43 | 0.39 |
| 2 | 1.03 | 124 | 1.06 | 262.05 | 0.99 | 0.65 |
| 3 | 1.02 | 103 | 1.04 | 324.82 | 1.03 | 0.47 |
| 4 | 1.02 | 210 | 0.93 | 734.02 | 1.38 | 0.22 |
| 5 | 0.97 | 158 | 0.98 | 409.89 | 1.05 | 0.36 |
| 6 | 0.84 | 101 | 0.91 | 225.45 | 1.41 | 0.27 |
| 7 | 1.05 | 208 | 1.06 | 664.27 | 1.52 | 0.15 |
| 8 | 1.07 | 159 | 1.07 | 763.02 | 0.90 | 1.12 |

**D**

| Pocket | SiteScore | Size (Å) | DScore | Volume (A³) | Phobic (LogP) | Philic (LogP) |
|---|---|---|---|---|---|---|
| 1 | 1.05 | 193 | 0.93 | 288.15 | 1.46 | 0.44 |
| 2 | 0.99 | 140 | 1.01 | 260.42 | 1.03 | 0.48 |
| 3 | 1.03 | 109 | 0.90 | 321.92 | 1.49 | 0.06 |
| 5 | 1.01 | 222 | 0.96 | 593.02 | 1.24 | 0.39 |
| 7 | 0.99 | 214 | 0.97 | 693.24 | 1.15 | 0.43 |
| 8 | 0.98 | 208 | 1.03 | 720.35 | 1.03 | 0.62 |

**Fig 2. Potential orthosteric and allosteric pockets in PfIleRS and HsIleRS, and their predicted druggability attributes.** (A) PfIleRS orthosteric and allosteric pockets. (B) HsIleRS orthosteric and allosteric pockets. (C) Table of quality assessment parameters for each predicted pocket in PfIleRS: SiteScore, Size, DScore, Volume, Hydrophobicity, and Hydrophilicity. (D) Table of quality assessment parameters for each predicted pocket in HsIleRS.

terms of polarity, size, charge and hydrophobicity. S4 Table indicates that PfIleRS has high hydrophobicity compared to HsIleRS, while the pocket size in PfIleRS was structurally smaller than in HsIleRS. These dissimilarities favour the formation of bond interactions that are unique and specific to PfIleRS. The potential to specifically target PfIleRS with mitomycin via its editing domain pocket over HsIleRS has been previously exploited [7].

### 3.2. *Accurate and reliable blind-docking parameters are established and validated*

As we reported in our previous article [7], we again followed two steps of docking validation. First the AMPs in PfIleRS and HsIleRS redocked precisely into the nucleotide binding pockets with reference RMSDs of 0.413 Å and 1.307 Å respectively when compared to their original conformations in the modelled structures, using exhaustiveness of 700 (Fig 3A and B). As previously established, an RMSD value below 2 Å indicates the validity of a docking method [77]. Further, the redocked AMP had similar interaction profiles with the modelled AMP and common interacting residues include Gln746, Gln742, and Phe132 for PfIleRS, and Gln551, Gln555, and Phe41 for HsIleRS (S3 Fig).

In the second validation step, benchmarking analysis of the docking protocol using 11 active inhibitors (S5 Table) showed that by screening just 10% of the known bioactive compound library against PfIleRS, it was able to identify true positives early at an efficient rate of 98% (Fig 3C). The estimated Area Under the Curve (AUC) was 0.95 with a Boltzmann-Enhanced Discrimination on Receiver Operating Curves (BEDROC) of 0.76 which altogether indicates that the virtual screening protocol was reliable considering the accurate recognition of the known bioactives (Fig 3D). These findings closely agree with a previous PfIleRS docking validation report although with a different exhaustiveness [7]. Docking method validation has also been reported for other studies that screened available databases such as ZINC, PubChem, SANCDB, and DrugBank [7,78,79].

### 3.3. *Eleven SANCDB compounds selectively bind to PfIleRS*

SANCDB compounds were previously screened against drug targets for both infectious and non-infectious diseases [11,80–89]. Here, 1019 SANCDB compounds from the updated database [30] were, first, screened against PfIleRS and then against HsIleRS in a systematic manner. The docking and compound selection pipeline used in this study is shown in Fig 4. After the first round of blind docking against PfIleRS (Fig 4A), an in-house Python script was used to select compounds that docked to PfIleRS potential allosteric pockets. The script determines the pocket and ligand centroids by averaging pocket residues and atom coordinates respectively. The pocket and ligand centroid coordinates are then converted into matrices and a distance between the two points calculated. A cutoff distance of 10 Å was used to extract ligands in their respective pockets resulting in 855 compounds. Additionally, 164 compounds were in the active site of the protein and these were discarded. 855 compounds were further docked to HsIleRS protein of which 140 compounds docked to the active site of HsIleRS (Fig 4B). These were also discarded. Notably, two compounds, SANC140 and SANC968, were only docked to PfIleRS but not to HsIleRS. 713 compounds (855 - 140 - 2 = 713) that docked to HsIleRS potential allosteric pockets were re-docked to PfIleRS to confirm their binding pockets (Fig 4C). As a final step, a scatter plot of predicted binding affinities of HsIleRS against PfIleRS was used to identify compounds with strong affinity towards PfIleRS (Fig 4D).

Overall, we identified nine potential hit compounds (SANC382, SANC456, SANC522, SANC806, SANC1082, SANC1095, SANC1104, SANC1122, and SANC1126) that exhibit stronger binding affinity towards PfIleRS compared to HsIleRS. Two-dimensional presentation of these compounds is presented in Fig 4E. All identified potential allosteric modulators had binding affinities ≤ -8.1 kcal/mol, as shown in the bar-graph distribution (Fig 4F). Potential Absorption, Distribution, Metabolism, Excretion and Toxicity (ADMET) show that selected compounds satisfy all the drug-like properties as assessed by SwissADME webserver (S6 Table). Further, all 11 SANCDB compounds passed the Pan-assay Interference Compounds (PAINS) filter which suggests that our ligands have no false positive hits that could have interacted non-specifically with PfIleRS and HsIleRS targets. SANC382, SANC522, SANC806, SANC968, SANC1095, SANC1126 also showed potential drug lead-like properties (S6 Table).

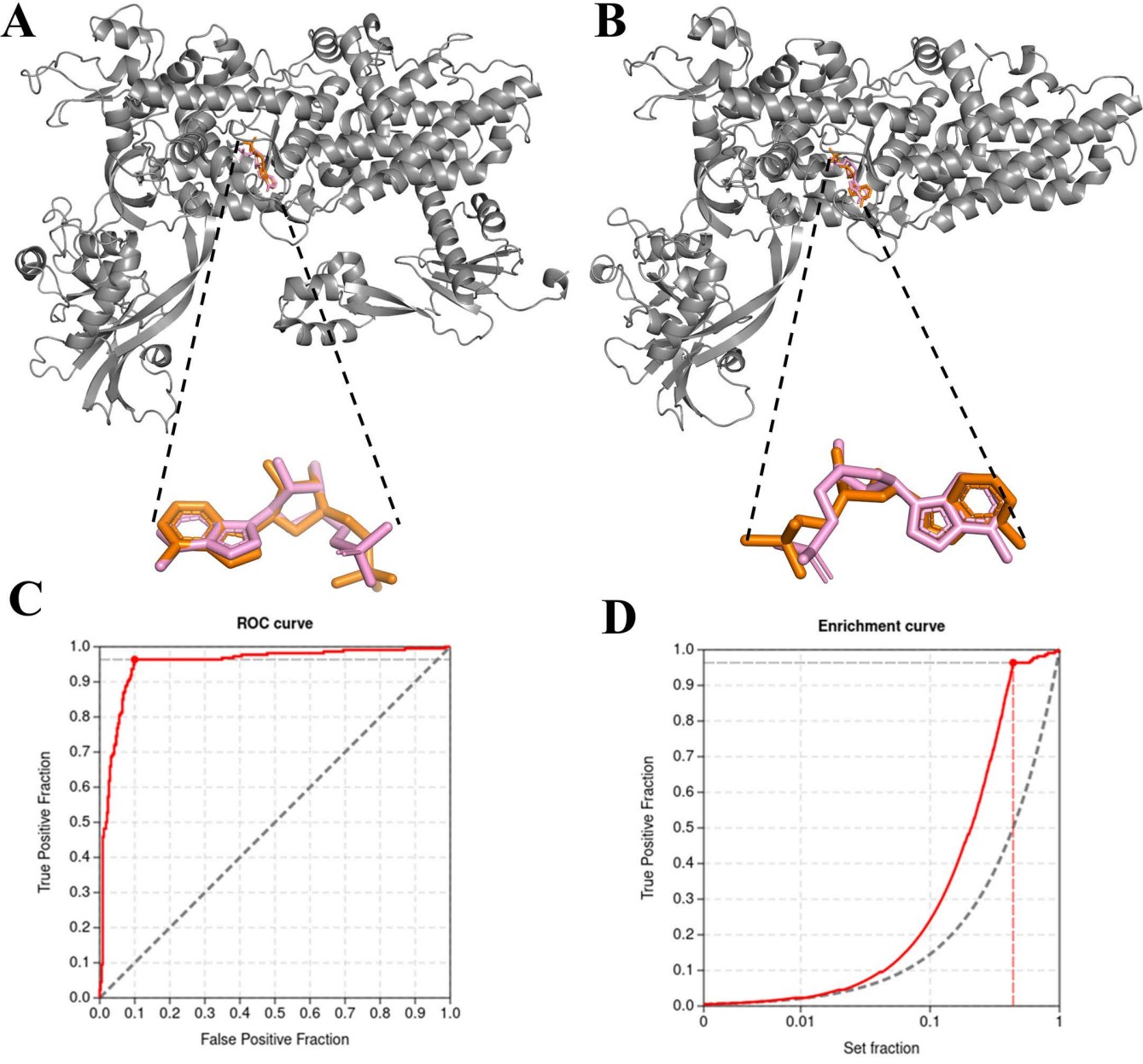

**Fig 3. Validation and calibration of AutoDock Vina docking parameters.** (A) Redocking validation of AMP to PfIleRS (orange) aligned with modelled AMP (pink). (B) Redocking validation of HsIleRS using AMP (orange) aligned with modelled AMP (pink). (C) ROC curve used for docking parameters calibration, indicating the optimal point where all active inhibitors were identified (red curve peak). (D) Enrichment Curve for docking active inhibitors and decoys to PfIleRS, achieving a BEDROC of 0.76 during AutoDock Vina calibration (red curve).

The **SANC140** compound (Discorhabdin A) is derived from marine sponges (*Latrunculia ssp*) and is known to be a potent antitumor alkaloid [90]. *In vitro* studies have shown that Discorhabdin compounds, including **SANC140**, to exhibit antimalarial activity against chloroquine-resistant strains of *Plasmodium falciparum* with low toxicity in mammalian cell lines, specifically monkey kidney fibroblasts [91]. **SANC968** (Thermospine), sourced from fruit barks and pods of *Sophora*

 

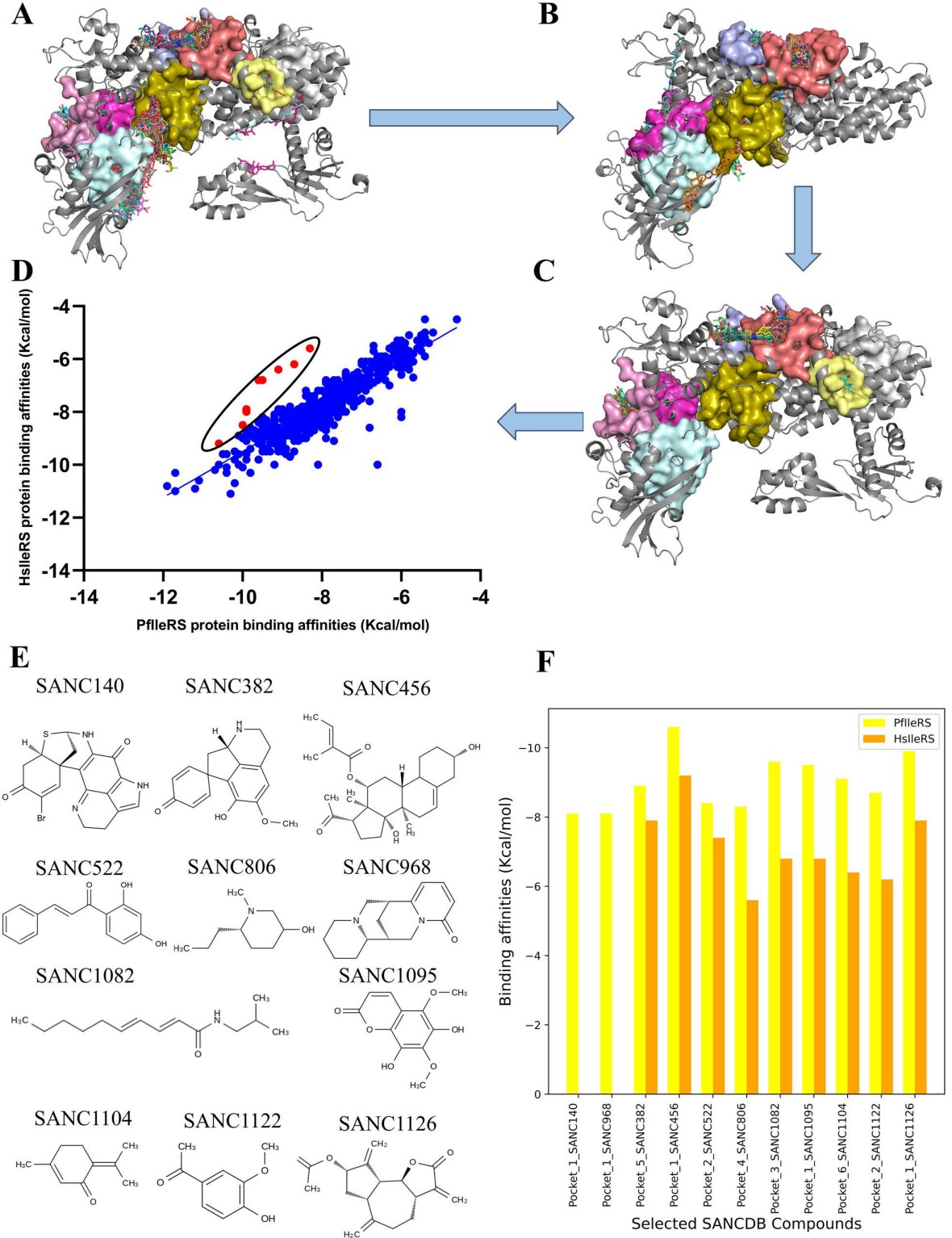

**Fig 4. SANCDB selection pipeline for identifying potential allosteric modulators of PfIleRS.** (A) Blind docking to PfIleRS holo protein. (B) Docking of extracted potential allosteric compounds to HsIleRS for selectivity evaluation. (C) Redocking of selected compounds to PfIleRS to identify their binding pockets. (D) Selection criteria based on binding scores used to identify the nine potential hit compounds (coloured in red). (E) Two-dimensional

representation of SANCDB potential hits against PfIleRS. (F) Bar graph showing the distribution of binding affinities of selected compounds in PfIleRS and HsIleRS. The binding pockets indicated in the bar are for PfIleRS.

*velutina ssp*, possesses antibacterial properties, although its activity as an antiplasmodial agent has not been reported [92].

**SANC382** (Crotsparine), isolated from *Croton Bonplandianum baill*, has been documented to synergize with Sparsiflorine against *Pseudomonas aeruginosa*, a common nosocomial pathogen [93]. Crotsparine has demonstrated antiplasmodial activity against chloroquine-sensitive and chloroquine-resistant strains of *Plasmodium falciparum* [94]. Additionally, **SANC456** (Gordonoside A), previously identified as a modulator for PfProRS protein [11], is an appetite suppressant found in Hoodia species [95]. **SANC522** (2',4'-Dihydroxy-chalcone) exhibits inhibitory effects against HSP90 and has antifungal properties [96]. Chalcones are known to target plasmodial cysteine proteases used by the parasite for haemoglobin hydrolysis [97]. **SANC806** (N-Methyl Pseudoephedrine) is a marker impurity of methamphetamine, crucial in forensic investigations [98]. Pseudoephedrine, a sympathomimetic drug related to ephedrine, is available over the counter as a decongestant and anorectic agent [99]. Furthermore, **SANC1082** (Pellitorine) is a potential anticancer compound isolated from *Piper nigrum* [100]. *In vitro* assays have shown Pellitorine has antiparasitic activity with IC50 values within a range of (0.05–311 µM) [101]. Regarding **SANC1095** (6,8-Dihydroxy-5,7-dimethoxycoumarin), this potential hit exhibits cytotoxic activity against L1210 cells in vitro [102]. Coumarin derivatives from *Artemisia apiacea* are crucial components in the development of artemisinin [103]. **SANC1104** (Diperiteone) is reported to be toxic and retards malaria vectors such as *Anopheles stephensi* [104]. **SANC1122** (Acetovanillone), also known as apocynin, is utilised as a specific inhibitor of NADPH oxidase [105]. Acetovanillone exhibits various pharmacological properties, including anti-inflammatory, antimalarial, antifungal, and anticancer activities [106]. Finally, **SANC1126** (Zaluzanin D), a sesquiterpene lactose from *Vernonia arborea* leaves, has shown significant medicinal value with anti-inflammatory properties [107]. Zaluzanin D is reported to possess insect antifeedant, anti-tumor, and antifungal activities [108].

### 3.4. *SANCDB compounds bind to distinct allosteric pockets in PfIleRS and HsIleRS with varied interactions*

From our results, the binding of the SANCDB compounds to PfIleRS was mediated by alkyl, pi-alkyl, and pi-cation interactions, conventional hydrogen bonds, and van der Waals forces. Fig 5 provides a detailed view of the binding interactions of PfIleRS pocket 1 compounds, while S4 Fig presents the interactions of all 11 compounds. The hydrophobic nature of allosteric pockets coupled with the aromatic features of the compounds seemingly favoured the formation of aromatic (pi) interactions. Most SANCDB compounds identified have benzene rings in their scaffolds, an important characteristic in drug-like molecules [109]. This explains why alkyl, pi-alkyl, and pi-cation hydrophobic interactions were dominant between PfIleRS and the identified potential hits. Alkyl and pi-alkyl are interactions between a pi-electron cloud over an aromatic group or any alkyl group [110].

Specifically, **SANC140** formed a conventional hydrogen bond with residue Lys228 and a pi-alkyl interaction with residue Trp224. **SANC968** interacted with PfIleRS through pi-alkyl bonds involving residues Phe111, Trp224, and Val215. Both **SANC140** and **SANC968** are located in the PfIleRS pocket 1 which corresponds with a well-defined pocket at the Rossman fold region of the protein as earlier discussed in Section 3.1. **SANC456**, **SANC1095**, and **SANC1126** also docked to the same pocket 1 similar to **SANC140** and **SANC968** and characterised by residues Ile101→N115 and Val215→N229 (Fig 2A and S2 Table). Ligand interaction profiling shows that **SANC456** interacted with PfIleRS through a hydrogen bond at residue Ile225 and pi-alkyl bonds at residues Leu102, Trp105, Phe112, and Trp224. **SANC1095** formed two hydrogen bonds with PfIleRS at residues Arg223 and Trp224, and alkyl bonds at residues Leu102, Trp105, Val215, Val218, and Phe227. **SANC1126** formed alkyl interactions at residues Phe111, Val215 and Phe227 with PfIleRS protein.

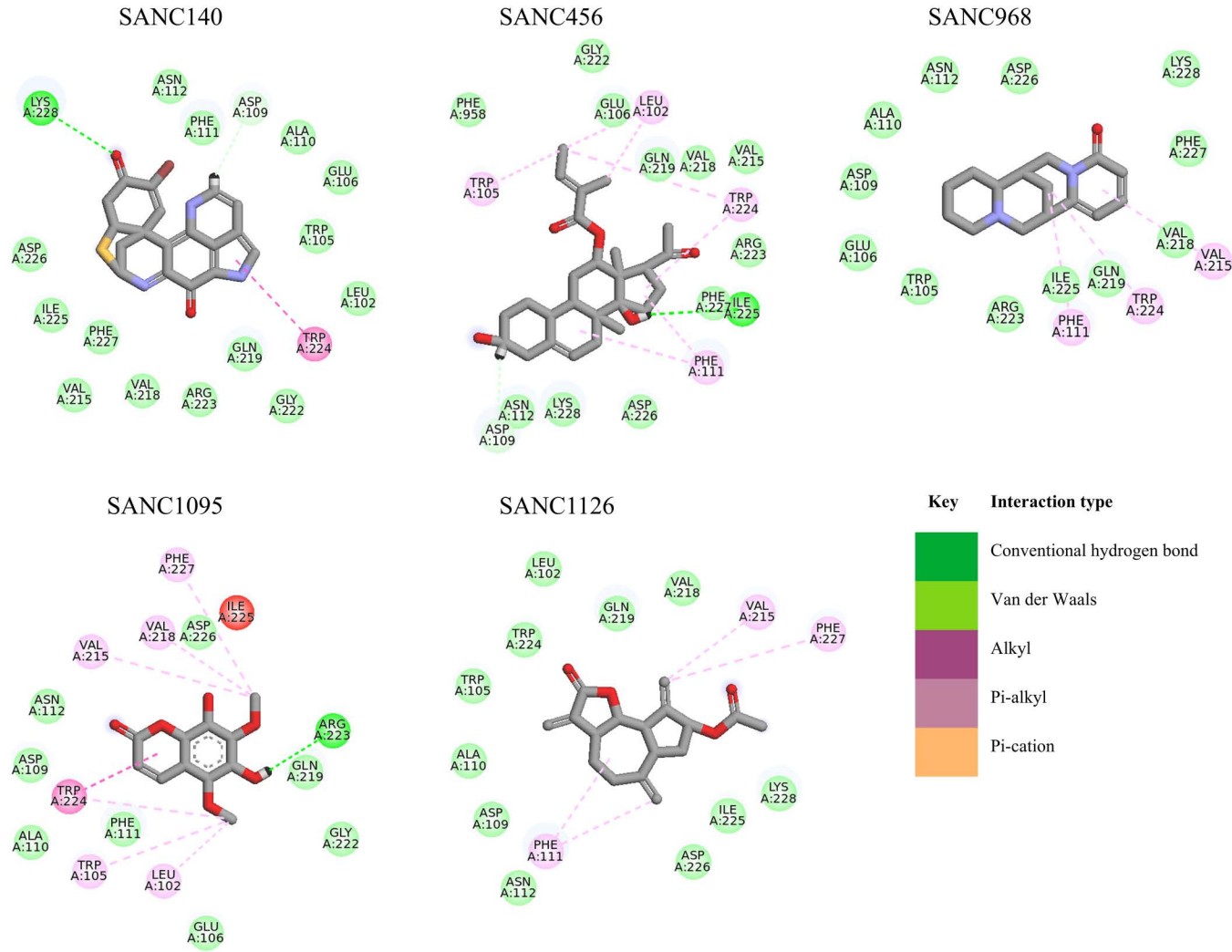

**Fig 5. Protein-ligand interactions of PfIleRS allosteric pocket 1 potential SANCDB hit compounds.** All possible interactions formed between the PfIleRS pocket 1 and the SANCDB compounds are shown with their respective binding modes.

**SANC522** and **SANC1122**, on the other hand, were bound to potential allosteric pocket 2 proximal to pocket 1 (at the Rossman fold region), and it is defined by residues K119→N128, Y161→R165, and M715→I734 (S4 Fig and S2 Table). In addition to hydrogen bonds, a pi-cation bond was involved in the binding of **SANC522**, and **SANC1122** with residues Glu163, Phe259, Asp260, and Asp633. Pi-cation interactions occur when negatively charged amino acids interact with electron-rich benzene rings or amine side chains, stabilising the protein-ligand complex [109]. Furthermore, pi-alkyl bonds were involved at residues Pro123, Ile734, and Ile175 in the **PfIleRS-SANC522** interaction, while alkyl bonds were present at residues Ile125, Ile734, and Pro123 in the **PfIleRS-SANC1122** interaction.

**SANC1082**, **SANC806**, **SANC382**, and **SANC1104** were bound to allosteric pockets 3, 4, 5, and 6, respectively (S4 Fig). Potential allosteric pocket 3 was mapped by residues F863→N884, N939→L947, and N977→N981, while potential allosteric pocket 4 located at the CP2/3 intersection was defined by residues Y256→V261 and P504→D529, and potential allosteric pocket 5 was mapped by residue E508→R512 and Y588→C635, and finally, allosteric pocket 6 was located around residues N93→E98, I836→Y840 and I959→I963 many of which contributed to the binding of the

SANC compounds. Interestingly, some of the residues from allosteric pockets 4 and 5 such as Y506 and D509 have been previously reported for their key roles in the binding of mitomycin at the PfIleRS editing domain [7]. The interactions of **SANC1082** with residues at its target site in the anticodon binding domain (pocket 3) were majorly mediated by pi-alkyl bonds. Pi-cation interactions with Phe259 and pi-alkyl interactions with Thr511, Leu527, and Ala590 characterised the binding of **SANC806** at the CP2/3 region (pocket 4). Moreover, the binding of **SANC382** at the CP2/3 region (pocket 5) involved pi-cation bonds with Lys260 and Asp633; conventional hydrogen bonds with Val261, Tyr256; pi-alkyl with Tyr511 and a van der Waals contribution by Pro263. **SANC1104** binding at the anticodon binding domain (Pocket 6) was mediated by alkyl bonds with PfIleRS at residues Ile94, Leu837, Ile959, and Trp840 and a strong conventional hydrogen bond with residue Tyr962.

It is worth noting that in the human homolog, these potential hits did not bind to the active pocket nor were they binding to the same potential allosteric pockets as reported for PfIleRS as shown in S5 Fig. **SANC382**, **SANC1095,** and **SANC1126** were all bound to potential allosteric pocket 2 in the homolog HsIleRS which differs from their preferred binding pocket in the plasmodium protein. This predicted pocket was defined by residues Q21→D37, H70→R74, I134→F147, and A528→A543 (S3 Table). Thus, **SANC382** interacted with the HsIleRS protein via an unfavourable donor-donor bond at residue Tyr140, unfavourable acceptor-acceptor at residue Asp72 which may likely reduce the binding affinity. Two conventional hydrogen bonds were formed at residues Asp37, and Arg74. **SANC1095** interacted with HsIleRS forming three conventional hydrogen bonds at residues Phe35, Asp72, and Gln539. Also, pi-alkyl bonds were formed at residues Lys32 and Ala543. **SANC1126** interacted with HsIleRS forming one conventional hydrogen bond at residue Arg74 and one alkyl bond at residue Ala543. **SANC456** and **SANC522** which binds in the Rossman fold region in PfIleRS were positioned in the potential allosteric pocket 7 (editing domain) of HsIleRS defined by residues; V170→R254, Y304→D324, and F385→R445. **SANC456** formed only one conventional hydrogen bond at residue Arg254 and pi-alkyl at residue Val314. Three compounds; **SANC806** (PfIleRS – CP2/3), **SANC1082** (PfIleRS - ABD), and **SANC1122** (PfIleRS – RF) were docked to potential allosteric pocket 3 defined by residues; P31→F69, D547→I580, and G695→V799. Alkyl bonds were formed at Tyr63, Ala64, and Met701 residues. A conventional hydrogen bond was formed at Ser67 residue as a result of the **HsIleRS-SANC806** interaction. For **HsIleRS-SANC1082**, only pi-alkyl bonds were formed at Ile59, Val60, Tyr63, Tyr704, and Leu706 residues. Only **SANC1104** was binding to potential allosteric pocket 5 which is located at the CP2/3 region of HsIleRS but the ABD region in PfIleRS. Interactions mediated by the ligand in HsIleRS include a hydrogen bond at Val170 and alkyl bonds with Tyr319, Tyr325, Pro340, and Ala403. These differential binding mechanisms, altogether, could account for the high binding affinity exhibited by these selected 11 SANCDB hits towards PfIleRS over HsIleRS, as obtained from the docking analysis thereby indicating the selectivity potentials. The differential binding stabilities of these compounds were further explored via MD simulations as explained in Section 3.5.

### 3.5. *SANCDB compounds modulate global flexibility and compactness of PfIleRS*

Due to the high flexibility of the C-terminal domain, it was removed from the structure prior to MD simulations. Here, we calculated RMSDs for three holo-state Pf and Hs proteins and their protein-ligand complexes over 100 ns MD simulations per system. The results for entire trajectories are presented as line plots (S6 Fig) and for the last 80 ns (equilibrated sections in the holo systems as reference) in violin plots (Fig 6). Notable, while the three holo-protein replicates presented a consistent and stable RMSD profile, the introduction of the SANCDB compounds, particularly **SANC140**, **SANC456**, **SANC968,** and **SANC1095** altered protein dynamics (Fig 6A and S6 Fig). Interestingly, these compounds were binding to the Rossman fold region (pocket 1). Some of the protein-ligand complexes (**PfIleRS-SANC382** and **PfIleRS-SANC1122**) maintained unimodal RMSD distribution similar to the holo states, indicating minimal global conformational changes (Fig 6A). In contrast, other ligand-bound systems displayed bimodal distributions, suggesting potential allosteric modulations in the presence of these compounds (Fig 6A).

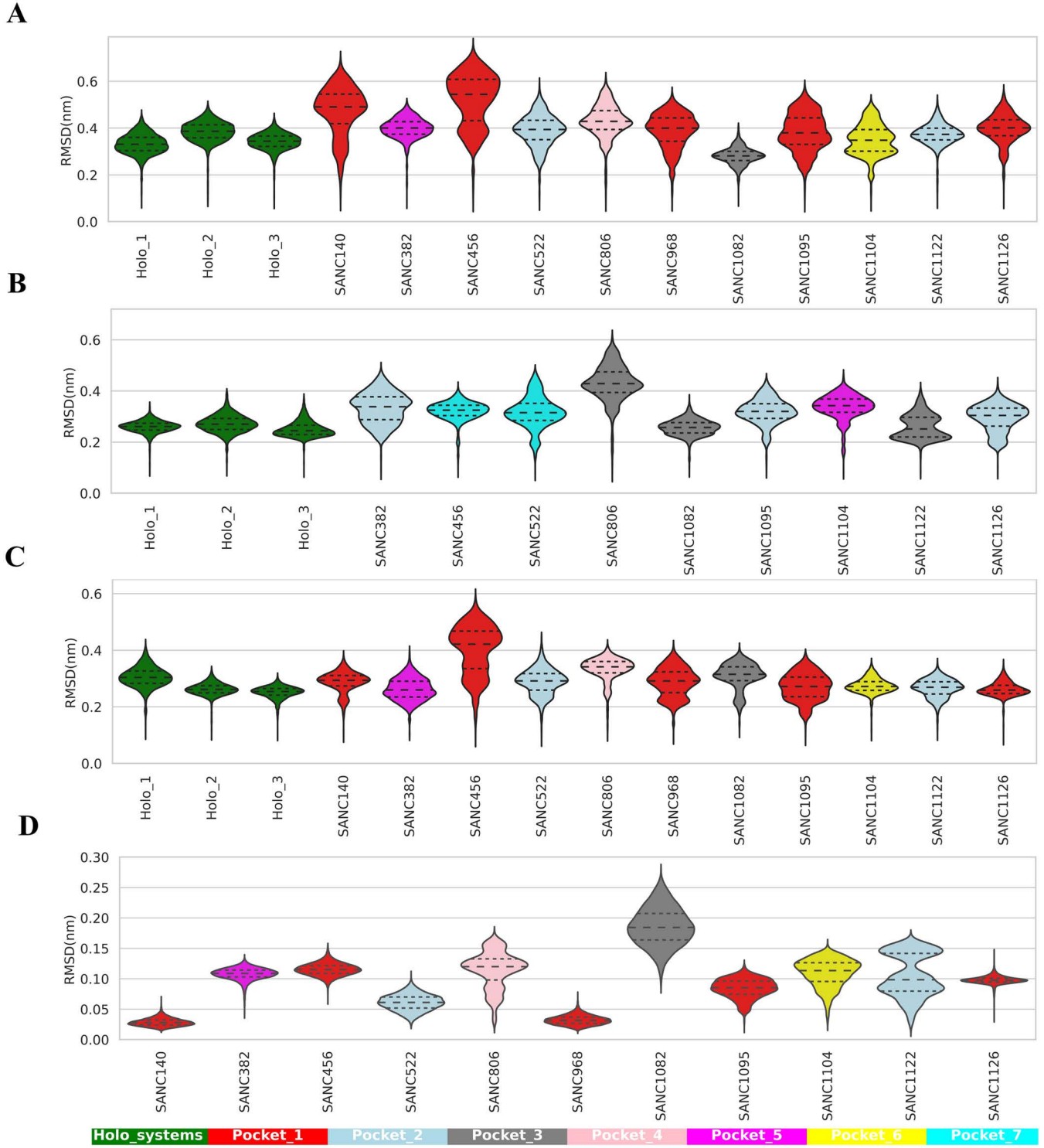

**Fig 6. Global trajectory analysis of PfIleRS and HsIleRS last 80 ns equilibrated portions.** (A) PfIleRS backbone-RMSD violin plots of the holo states and PfIleRS-SANC complexes. (B) HsIleRS backbone-RMSD violin plots for the holo and HsIleRS-SANC complexes. (C) Orthosteric pocket (masked residues defining the pocket) backbone-RMSD violin plots of holo and PfIleRS-SANC complexes. (D) Selected SANCDB compounds RMSD violin plots for 100 ns simulation.

In the same way, we also determined the potential allosteric effects of the compounds in human homolog. HsIleRS-SANC complexes generally showed higher backbone RMSD compared to the holo states (S6 Fig C and Fig 6B). Notably, unlike the effect of **SANC382** and **SANC1122** on the Pf protein, the violin plots of equilibrated trajectories for HsIleRS with **SANC382** and **SANC1122** displayed bimodal distributions. This suggests potential allosteric modulation in human protein, but not in PfIleRS (Fig 6A and B). The other compound with a bimodal effect was **SANC1126**.

The rest of the HsIleRS-SANC systems exhibited a unimodal distribution, suggesting minimal allosteric effects on the global protein structure. Distinctly, **HsIleRS-SANC456** showed a unimodal distribution, in contrast to **PfIleRS-SANC456**, where significant conformational changes were observed (Fig 6A and B). This finding suggests that **SANC456** may selectively interact with PfIleRS over its HsIleRS homolog, hence potentially a good compound to develop further as a potential hit. Similarly, **HsIleRS-SANC1095** and **HsIleRS-SANC1104** displayed unimodal distributions, unlike the bimodal distributions observed in **PfIleRS-SANC1095** and **PfIleRS-SANC1104** complexes (Fig 6A and B). This indicates that these compounds binding at the Rossman fold region may exhibit selectivity towards PfIleRS.

As a next step, we investigated if these potential allosteric modulators also have effects on active site pocket conformation. RMSD violin plots were specifically calculated for the active site pocket per protein system using residues defining these pockets as identified by pocket prediction programs (S2 and S3 Tables). Active site pockets of PfIleRS-SANC complexes, interestingly, demonstrated a similar trend to their respective entire protein-compound RMSD profile (Fig 6C). Averaged mean RMSD values were calculated for all eleven PfIleRS-SANC systems to estimate their deviation from the holo references (S7 Table). The highest mean RMSD deviations were observed for some PfIleRS RF-binding compounds: **SANC140** and **SANC456**, agreeing with the global RMSD line and violin plots (Fig 6A and S6 Fig A). Interestingly, although the **HsIleRS-SANC456** complex had the highest deviation compared to the holo states, it was significantly lower than the averaged RMSD value in the **PfIleRS-SANC456** system (S7 and S8 Tables).

As a third step, we further analysed the trajectories by clustering them for the last 80 ns. Clustering assembles frames with the same conformation using a partition algorithm, grouping them into clusters. Most of the PfIleRS-SANC complexes showed two clusters (S9 Table). Only **PfIleRS-SANC522** and **PfIleRS-SANC1082** remained primarily in one geometric cluster throughout the 80 ns trajectory. These clustering results align with the backbone RMSD results, which showed a bimodal distribution for the global confirmation of most PfIleRS-SANC systems. Specifically, clustering revealed major conformational changes at the editing and junctional domains of the PfIleRS-SANC complexes (S7 Fig). The holo systems were compact in the editing and junctional domains in comparison to PfIleRS-SANC complexes. Notably, PfIleRS systems: **SANC140** (pocket 1), **SANC1095** (pocket 1), **SANC522** (pocket 2), and **SANC806** (pocket 4) have a secondary structure change of the two anti-parallel beta-sheets in the editing domain (Y281→I294 and M567→Y576) that link the domain to the rest of the Rossmann fold catalytic domain (S8 Fig). Importantly, these anti-parallel beta-sheets did not change their secondary structure in the holo systems.

Clustering of HsIleRS-SANC complexes to determine the most sampled conformations during MD simulations revealed that most structures were in two clusters, except for **SANC456**, **SANC1095**, and **SANC1104**, which were primarily in one conformation (S10 Table). This finding aligns with the HsIleRS-SANC RMSD violin plots, which showed bimodal distributions for **SANC382**, **SANC522**, **SANC806**, **SANC1122**, and **SANC1126** (Fig 6B). Unlike PfIleRS-SANC clusters, HsIleRS-SANC complexes exhibited less RMSD deviation across the three geometric clusters, although notable changes were observed in the editing domain (S9 Fig). Significant RMSD deviations were also observed in HsIleRS around the Rossmann fold due to the numerous loops linking alpha helices and beta-sheet regions [111].

Lastly, to assess the impact of allosteric modulation on PfIleRS, we checked the stability of potential hit compounds over the simulation using RMSD. The results showed that the compounds remained in their pockets throughout the simulation (Fig 6D). **SANC140**, **SANC456**, and **SANC968**, which targeted the PfIleRS RF region, significantly impacted the protein's global conformation as seen in global protein RMSD, and were stable throughout the trajectory (Fig 6D). **SANC456**, also known as 5α-Pregna-1,20-dien-3-one (Gordonoside A), was previously identified as a potential hit against

*Plasmodium falciparum* Prolyl tRNA Synthetase (PfProRS) [11]. Hypothetically, **SANC456** could be a multitarget compound, capable of modulating more than one aaRS drug target. **SANC806** and **SANC1122** in complex with PfIleRS had bimodal distribution. HsIleRS ligands were also stable, except for **SANC522**, **SANC1082**, and **SANC1122** which flipped during the simulation (S10 Fig). Overall, all selected potential hit compounds remained in their respective pockets throughout the simulation.

### 3.6. *Effect of allosteric modulation on PfIleRS residue flexibility*

The largest holo enzyme RMSF regions were between residues W400→A495 and D1012→L1091, located primarily in loop regions of the editing domain and junctional domain, respectively (Fig 7A). ABD residues N981→I989 also had large RMSF; this region is close to high RMSF region D1012→K1014 (junctional domain) and likely interacts with one another. In HsIleRS, high atomic residue fluctuations were also seen in the loops of the editing and anticodon binding domains, as well as the loops of other regions (S11 Fig).

   The delta RMSF heatmap shows the RMSF differences of all systems compared to the holo average RMSF (Fig 7B). As expected, the major inconsistent delta regions of the three holo replicates correspond to loopy high initial RMSF regions, namely residues P174→D190 (NT-RF), L406→V452 (editing domain), E979→D993 (ABD), I1017→D1030 (JD), and N1059→R1068 (JD). These regions in the SANC systems are analysed with more caution. Notably, junctional domain residues R1036→T1054 are consistent in the holo replicates but have increased RMSF in all but two SANC systems (Fig 7B and S12 Fig), increasing up to 5 Å from the holo. Interestingly, all pocket 1 systems showed an increase here. Also consistent in the holo replicates, the C-terminal end of the junctional domain (K1074→L1091) has large RMSF changes in several SANC systems. Although the replicate holo editing domain RMSF was inconsistent (L406→V452), **PfIleRS-SANC456** showed considerable increased RMSF of this region while **PfIleRS-SANC1082**, **PfIleRS-SANC806**, and **PfIleRS-SANC382** showed widespread decreased RMSF (Fig 7B and S12 Fig). Residues I1017→D1030 (JD) were also inconsistent in the holo systems, however, all PfIleRS-SANC systems showed decreased RMSF here; this may be important as this is the link between the anticodon binding domain and the junctional domain. Lastly, large RMSF changes near the active site were explored, all of which were RMSF increases. Residues D230 and Y231 had increased RMSF in **PfIleRS-SANC1122** and **PfIleRS-SANC1082**. Residues L277→N280 had increased RMSF in **PfIleRS-SANC456**, **PfIleRS-SANC1095**, **PfIleRS-SANC522**, **PfIleRS-SANC1122**, and **PfIleRS-SANC1104**. Residues W580 and R581 (part of CWRS ZnF2) had increased RMSF in **PfIleRS-SANC456**, **PfIleRS-SANC1095**, and **PfIleRS-SANC522**. Residues R748 or G749 had increased RMSF in **PfIleRS-SANC140**, **PfIleRS-SANC456**, **PfIleRS-SANC968**, **PfIleRS-SANC1095**, and **PfIleRS-SANC522**. Residues S785→Y791 (part of KMSKR motif) had increased RMSF in **PfIleRS-SANC456,** including other RF-binding systems such as **PfIleRS-SANC140**, **PfIleRS-SANC1095**, and **PfIleRS-SANC522**. Interestingly, previous studies have revealed that the disruption of critical residues on the KMSKR motif, particularly K786, can severely affect the enzymatic activity of the protein [112–115].

### 3.7. *Compound-induced allostery disrupts the binding affinity of catalytic AMP*

Considering the importance of the nucleotide molecule in the aminoacylation reaction step, we evaluated the tendency of the identified SANCDB hits to indirectly (or allosterically) impact or modulate its binding affinity or mechanism. To this effect, we first compared the net binding free energies ($\Delta G^{bind}$) of the AMP molecules across the holo and ligand-bound systems. The three holo systems had AMP $\Delta G^{bind}$ values of -82.8 Kcal/mol, -86.7Kcal/mol, and -79.7 Kcal/mol (Fig 8A and S11 Table), which were increased by at least 25% in all SANC ligand-bound systems (Fig 8A). This analysis also revealed ligand-induced changes in $\Delta E^{vdW}$ and $\Delta E^{ele}$ which appear to be critical contributors to the binding of AMP (S11 Table), altogether implying that the SANC ligands affected the binding of the nucleotide molecule at the catalytic site. The large AMP binding affinity reduction of RF-binding compounds **SANC140**, **SANC456**, **SANC968**, and **SANC1095** could be related to their destabilising effect on the global conformation of the protein as discussed in Section 3.5.

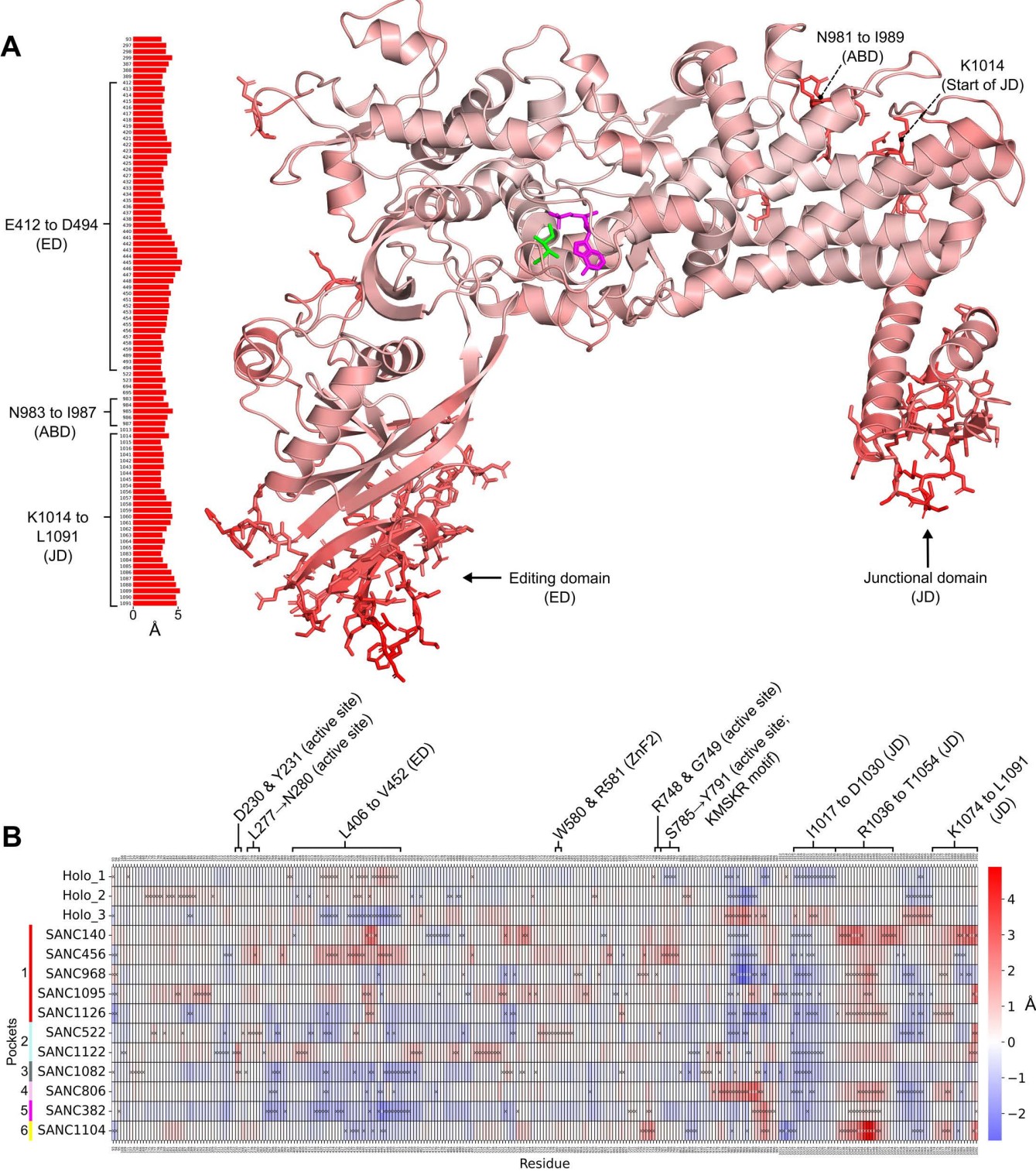

**Fig 7. RMSF.** A) Holo average RMSF values mapped to structure, with largest RMSF residues (top 10%: 91 residues) shown as sticks and shown in bar plot. AMP and Ile ligands are coloured magenta and green, respectively. The largest RMSF regions are located primarily in loop regions of the editing and junctional domains. B) Delta RMSF heatmap. The blue colour indicates a decrease from the holo-averaged protein, while the red colour indicates an increase. Residues per system with the largest change in RMSF (top/bottom 3% delta: 54 residues) relative to holo are shown with an "x". Residues

in the heatmap not shown with an "x" are not in the top/bottom 3% in their respective system; these are included for comparison purposes. Junctional domain residues R1036→T1054 are consistent in the holo replicates but have increased RMSF in all but two SANC systems. The largest active site RMSF change was seen in PfIleRS-SANC456, and then PfIleRS-SANC522.

To further understand the underlying mechanisms, changes in the net total energy ($\Delta E^{tot}$) contributions of key residues that directly contributed to the binding of AMP at the catalytic site were evaluated by analysing the per-residue energy contributions. This helped map out residues that are proximal to or in direct contact with AMP such as the highly conserved HYGH and KMSKR residues (Fig 8B). From our calculations for the averaged holo systems, P131, F132, P137, H138, G140, H141, L143, A144, W214, R581, E742, G743, Q746, L775, V776, K783, S785, and K786 largely contribute

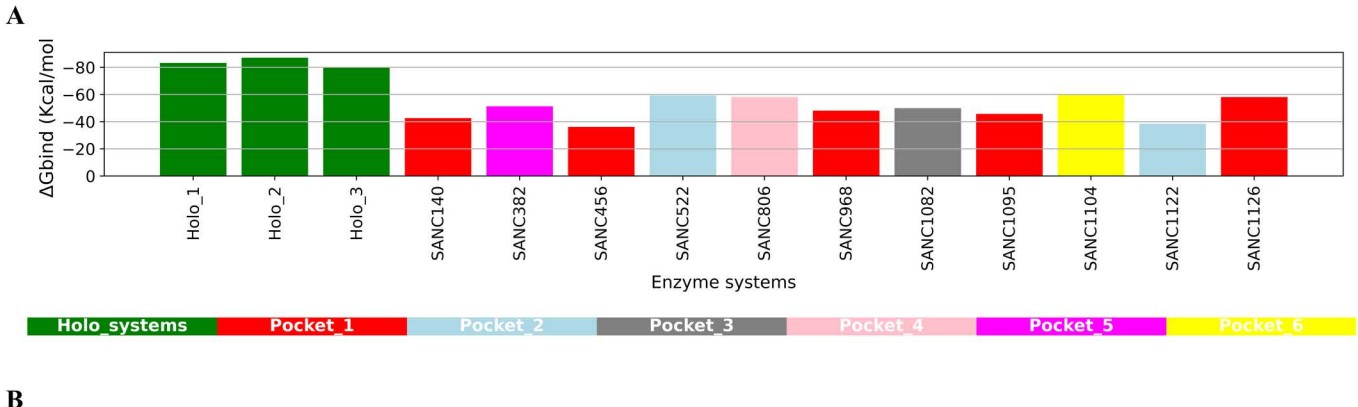

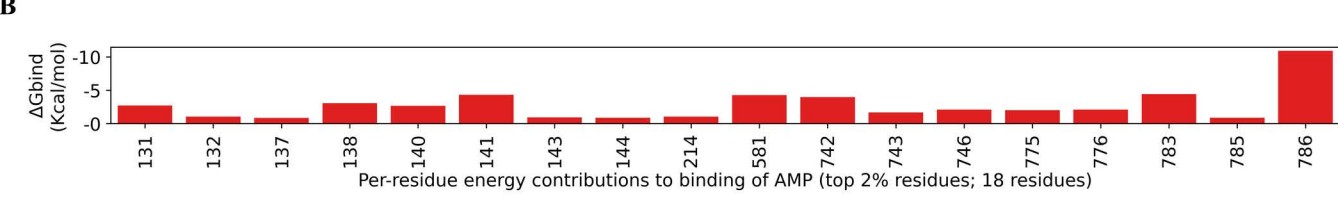

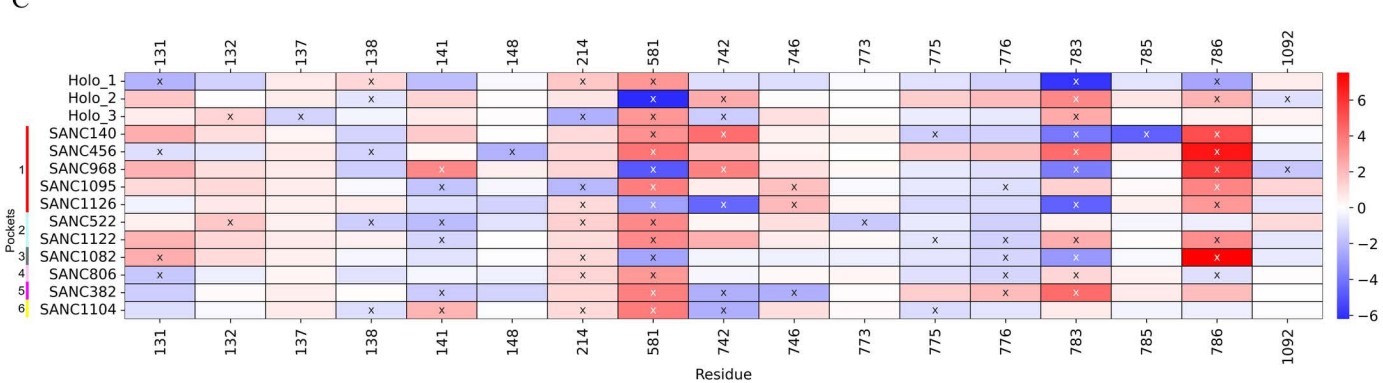

**Fig 8. AMP binding free energy calculations and per-residue decomposition of holo and PfIleRS-SANC systems.** (A) AMP energy changes in the overall $\Delta G^{bind}$ in the three holo states and ligand-bound systems upon allosteric modulation. (B) The holo-average top 2% residues (18 residues) contributing to the binding of AMP (<0.5Kcal/mol). (C) Delta per-residue decomposition of residues contributing to AMP binding in ligand-bound systems vs holo. Relative to the holo system, residues per system with the largest change in $\Delta G^{bind}$ (top/bottom 3 residues) are shown with an "x". Residues in the heatmap not shown with an "x" are not in the top/bottom 3 residues in their respective system; these are included for comparison purposes. The blue colour indicates a decrease from the holo-averaged protein, while the red colour indicates an increase.

to the binding of AMP above a set energy cut-off value of -0.5 Kcal/mol. Residue K786 has by far the highest contribution (-10.0 Kcal/mol) which is a part of the conserved catalytic KMSKR motif, alongside K783 ($\Delta E^{tot}$ = -4.0 Kcal/mol), and plays an important role in stabilising the active site conformation (Fig 8B). A decrease in $\Delta E^{tot}$ contribution however occurred in some of these key AMP-interacting residues in the presence of the SANC ligands which could explain the reduction in overall $\Delta G^{bind}$ values as earlier observed in the ligand-bound systems (Fig 8C). Of interest was residue K786 which showed a reduction in $\Delta E^{tot}$ contributions across many of the ligand-bound systems, particularly the RF-bound compounds (**SANC140**, **SANC456**, **SANC968**, **SANC1095**, and **SANC1126)**, which could further explain their reductive effects on AMP binding affinity. The **PfIleRS-SANC1122** and **PfIleRS-SANC1082** systems also had a reduction in AMP $\Delta E^{tot}$ contributions for K786 (Fig 8C). Other residues in systems with considerably decreased binding affinity were residue H141 in the **PfIleRS-SANC968** and **PfIleRS-SANC1104** systems, and residue Q746 in the **PfIleRS-SANC1095** and **PfIleRS-SANC1126** systems (Fig 8C). The effects of these compounds (particularly those that target the RF) on key residues of the KMSKR, such as K783 and K786 as observed here, could indicate their potential to allosterically alter PfIleRS enzymatic activity [115,116]. Complimentary to these events were the increases in CoM distances between both catalytic substrates Ile and AMP, more notably in RF-bound systems **PfIleRS-SANC456** and **PfIleRS-SANC1095**, and the ABD-bound **PfIleRS-SANC1104** system (S13 Fig). This could provide additional mechanistic evidence as to how the binding of the SANC ligands may allosterically destabilise the catalytic substrate molecules. This may also interconnect with the observed changes in the active site conformation for some of the ligand systems, for instance **PfIleRS-SANC456** and **PfIleRS-SANC1095** systems, which exhibited bimodal RMSD distributions (Fig 6C). Interestingly, **SANC456**, which is a RF-binding compound, showed the strongest binding among our SANC ligand hits ($\Delta G^{bind}$ = -34.9 Kcal/mol), interacted very stably with PfIleRS (unimodal RMSD distribution), and induced the most changes to the catalytic substrates. Other RF binding compounds: **SANC140**, **SANC522, SANC968**, **SANC1095,** and **SANC1126** also showed strong binding with $\Delta G^{bind}$ < -20.0 Kcal/mol which may have enhanced their modulatory effects on the PfIleRS global structure, active site, and catalytic substrates. These effects are also common for other compounds binding at other regions of the target protein such as the CP2/3 and the ABD (S12 Table).

### 3.8. *Dynamic residue network centrality elucidates the allosteric mechanism*

Network centrality measures the importance of a residue in terms of information flow/communication in the residue network [9,89,117]. *Betweenness centrality* (*BC*) quantifies the number of times a node acts as a bridge along the shortest path between two other nodes [66,89,118].

In Fig 9A the holo average *BC* values are mapped to the structure, with the top 5% residues (45 residues) shown as spheres on the structure, sticks in the enlarged window, and shown in a bar plot. Notably, the top 5% *BC* residues revealed the major allosteric pathway which spans the entire length of the enzyme, connecting the junctional domain to the active site and then the editing domain. From the junctional domain, the pathway traverses anticodon binding domain alpha-helices α30 and α31, then passes alongside the active site through residues V816, N773, G743, L744, D745, T747, and R748, then meets editing domain residues E276-Y281, H574-Y576, and V261-P263. Active site residues D745 and L279 are adjacent to each other and have by far the largest *BC* values. Surprisingly, of the important KMSKR (K783-R787), HYGH (H138-H141), and zinc finger motifs (S265-C268 and C579-S582), only one top 5% *BC* residue (R581; ZnF2) was found. The role of the high *BC* pathway may be to link the junctional and editing domains to the active site, while separately allowing the important active site motifs to perform their own function.

The pocket 1 systems, and pocket 2 system **PfIleRS-SANC522** (all RF-binding compounds), showed similar patterns of *BC* changes from holo (Fig 9B). Most systems here have decreased *BC* values (decreased inter-residue communication) in residues N278-Y281 (active site), S288 (editing domain), W395-T397 (editing domain), H574-I587 (editing domain/Zinc finger 2 hinge), and D745 (active site; largest holo *BC* residue). The decreased *BC* values of these important regions may be linked to the decreased AMP binding affinity. On the other hand, active site residues V261-F275 and G749 have

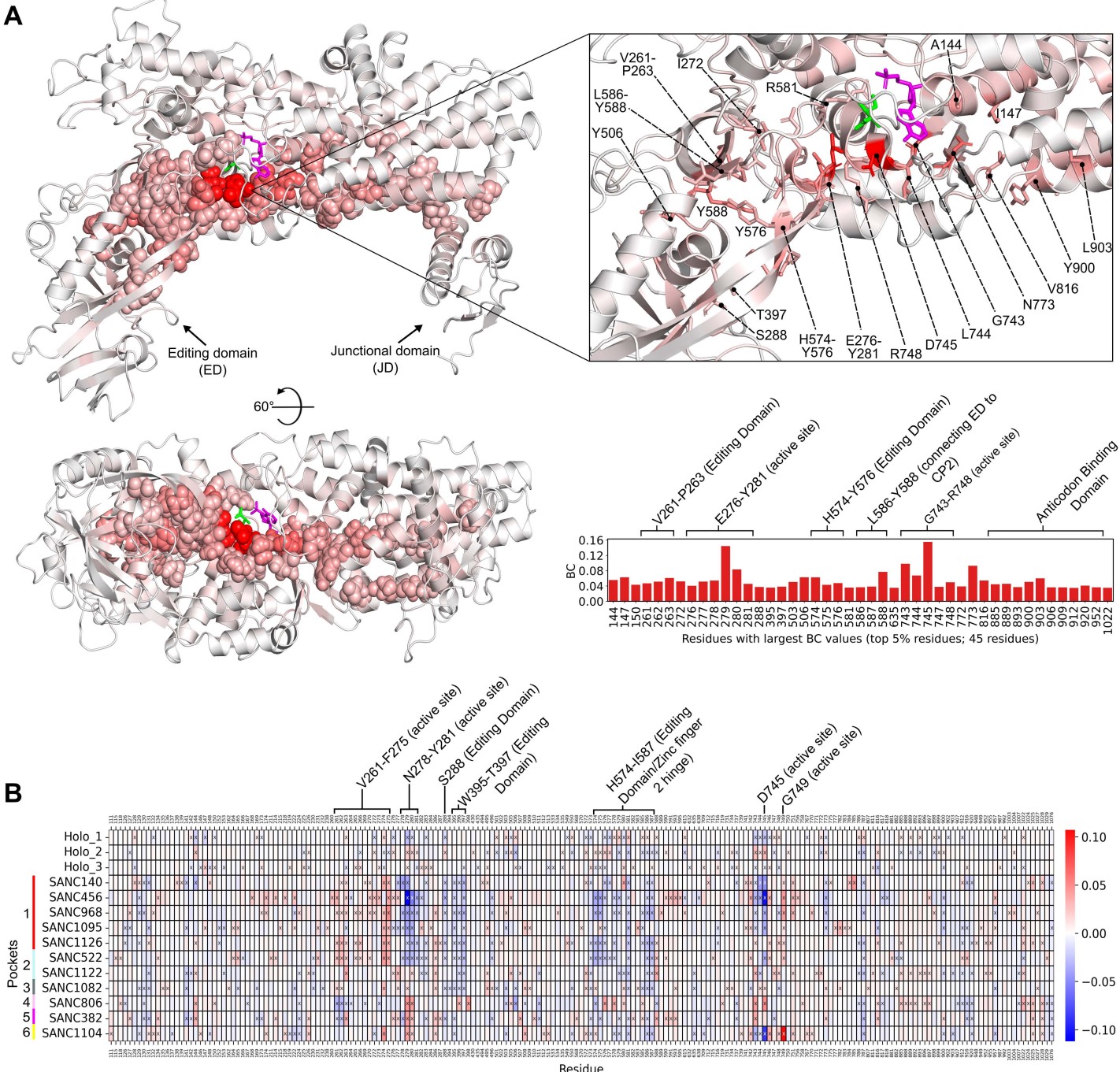

**Fig 9. BC.** A) Holo average *BC* values mapped to structure, with largest *BC* residues (top 5%: 45 residues) shown as spheres on the structure, sticks in the enlarged window, and shown in a bar plot. AMP and Ile ligands are coloured magenta and green, respectively. A major allosteric pathway is seen, spanning the entire length of the enzyme, connecting the junctional domain to the active site and then the editing domain. B) Delta *BC* heatmap. The blue colour indicates a decrease from the holo-averaged protein, while the red colour indicates an increase. Residues per system with the largest change in *BC* (top/bottom 3% delta: 54 residues) relative to holo are shown with an "x". Residues in the heatmaps not shown with an "x" are not in the top/bottom 3% in their respective system; these are included for comparison purposes. SANC ligand-induced *BC* changes of active site, zinc finger domain, and editing domain may be linked to enzyme functional change shown by the decreased binding affinity of AMP.

increased *BC* values. These delta *BC* residues are shown as spheres on representative structure **PfIleRS-SANC456** (S14 Fig A), with the remaining top/bottom 3% residues shown as sticks: A pathway of top 3% decreased delta *BC* residues can be seen between the active site and editing domain which may indicate decreased communication between the active site and editing domain.

The remaining systems have some similarities to the pocket 1 systems and **PfIleRS-SANC522** (pocket 2), but most residues show differing *BC* changes (Fig 9B and S15 Fig). For instance, **PfIleRS-SANC806** (pocket 4), **PfIleRS-SANC382** (pocket 5), and **PfIleRS-SANC1104** (pocket 6) have a large *BC* increase in residues L279-N280 (active site). **PfIleRS-SANC806** also has *BC* increases of residues T396, T398, Y576, F578, G743, and D745, while residues V261 and M262 have a *BC* decrease. Interestingly, the residues with increased *BC* values of **PfIleRS-SANC806** form a pathway starting in the editing domain where the **SANC806** ligand is situated, moving through the active site and towards the junctional domain through anticodon binding domain residues (S14 Fig B). This may affect the flow of information within the enzyme. The **PfIleRS-SANC382** system has *BC* increases in residues G507 and D745, while residues V261 and M262 have a *BC* decrease. Lastly, **PfIleRS-SANC1104** has the largest *BC* increase of any system in residue G749, and a large decrease of *BC* in residue D745 (Fig 9B).

Proteins often have allosteric communication pathways, where modulation can propagate signals that affect a distant functional pocket. These pathways are formed by residues connecting the two pockets [119]. Potential allosteric pocket 1 and 2 at the RF region is near the HYGH and KMSKR conserved motifs which are catalytic loops responsible for the orientation of the orthosteric pocket [120,121] and the ligand binding at this region could interfere with critical catalytic residue-residue communication. **PfIleRS-SANC456** (RF-binding compound) exhibited a pathway of top 3% increased *BC* residues starting at potential allosteric pocket 1 residues P957 and G222 and moving down residues V218, W214, W168, T134, and P174, affecting many other active site residues (S15 and S16 Figs). This could indicate allosteric modulation. On the other hand, RF-binding compound **SANC1095** induced large *BC* changes in residues R164, R165, Y231, and K232 (S16 Fig). **PfIleRS-SANC522** (RF pocket 2) exhibited a top 3% increased *BC* pathway of residues N115, S118, R164, R165, Y231, and M715 around the SANC ligand, which extended to top/bottom 3% delta *BC* residues D230, F166, W214, S211, and T134 in the active site (S14 and S15 Figs). Pocket 5 **PfIleRS-SANC382** exhibited top/bottom 3% delta *BC* residues Y511, G507, V261, A590, and I591 around the SANC ligand, leading to a top/bottom 3% delta *BC* residue potential allosteric path into the active site via residues M262, Y588, P263, L277, L586, N278, N280, L279, C579, and R581 (S15 and S16 Figs). As mentioned previously, the top 3% increased *BC* residues of **PfIleRS-SANC806** form a pathway starting in the editing domain where the **SANC806** ligand is situated (pocket 4), moving through the active site and towards the junctional domain through anticodon binding domain residues (S14 Fig B). Overall, these altered *BC* pathways may affect the flow of information within the enzyme and therefore enzyme function.

*Eigenvector centrality* (*EC*) measures how well connected a node is to other well-connected nodes in the network and, importantly, can identify separate functional groups/communities of nodes in a network [66]. In Fig 10A the holo average *EC* values are mapped to structure, with the top 5% residues (45 residues) shown as spheres on the structure, sticks in the enlarged window, and shown in a bar plot. There are two major groups/communities of residues, indicating extensive residue communication in these groups which may be important for enzyme function. The group on the left of the active site is mostly made up of CP3 domain residues (W642-T644 and V705-S714), while the group on the right contains important residues G140-C150 (active site; HYGH motif; α4), the entire alpha-helix 26 (Y801-I811), and some anticodon binding domain residues. These groups of high *EC* residues and their positions are likely important for active site stability and enzyme catalysis.

In the delta *EC* heatmap (Fig 10B), the most inconsistent regions of the holo replicates are found in regions with high holo average *EC* values, namely G167-I179 (NT-RF), N200-V207 (NT-RF), F238-F246 (NT-RF), S637-I646 (CP2 & CP3), and E704-S714 (CP3). These regions are located in the *EC* group on the left (Fig 10A). Although these regions have inconsistent holo replicate delta values, here most PfIleRS-SANC systems, particularly **SANC140**, **SANC456**, **SANC968**

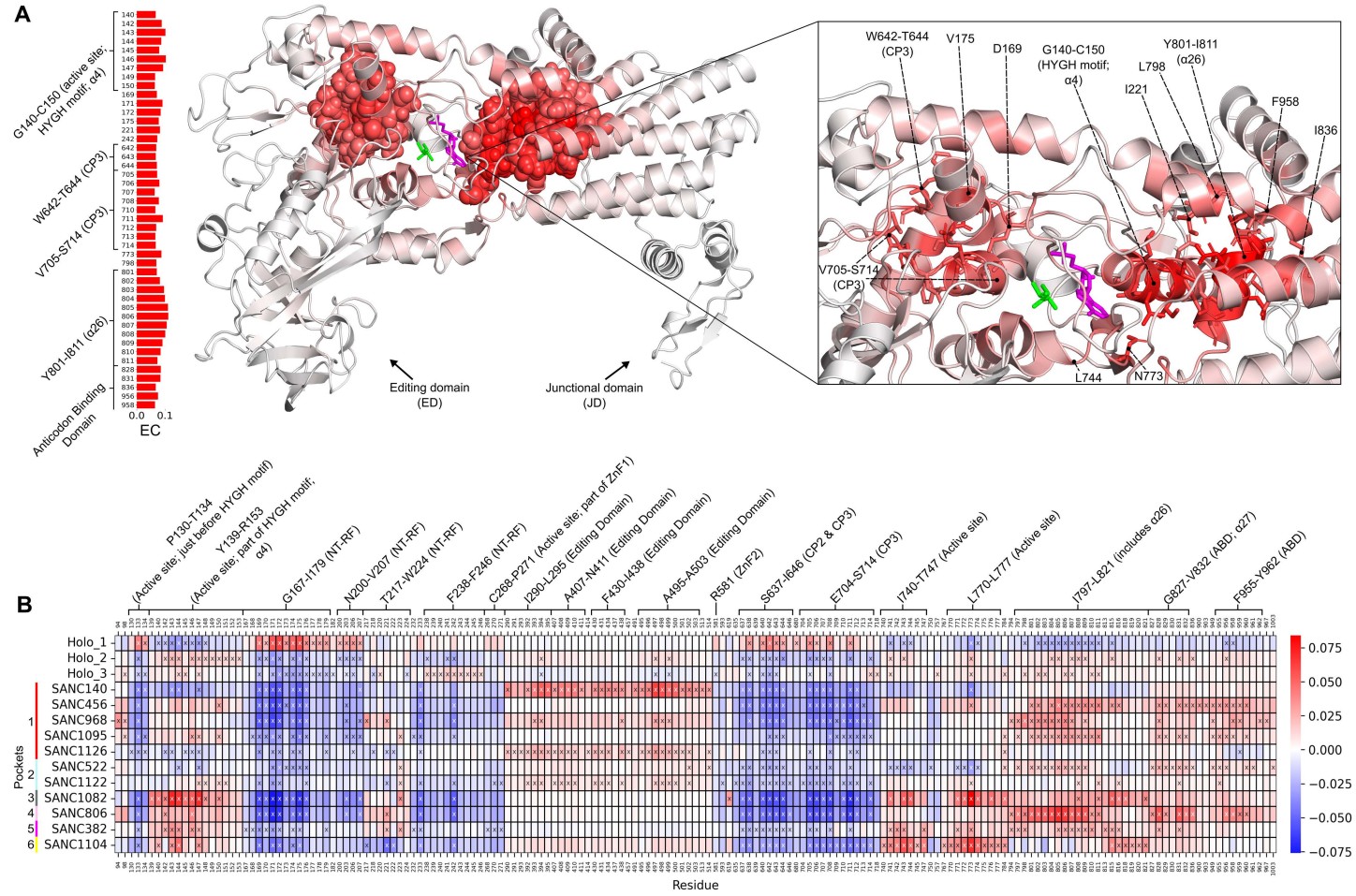

**Fig 10. *EC*.** A) Holo average *EC* values mapped to structure, with largest *EC* residues (top 5%: 45 residues) shown as spheres on the structure, sticks in the enlarged window, and shown in a bar plot. AMP and Ile ligands are coloured magenta and green, respectively. Two major groups/communities of residues are seen, indicating extensive residue communication in these groups which may be important for enzyme function. B) Delta *EC* heatmap. The blue colour indicates a decrease from the holo-averaged protein, while the red colour indicates an increase. Residues per system with the largest change in *EC* (top/bottom 3% delta: 54 residues) relative to holo are shown with an "x". Residues in the heatmap not shown with an "x" are not in the top/bottom 3% in their respective system; these are included for comparison purposes. In most PfIleRS-SANC systems, the high *EC* active site group seen on the left in Fig 10A exhibited decreased *EC* while the high *EC* active site group on the right exhibited increased *EC*.

**SANC1082**, **SANC806**, and **SANC1104**, have considerably decreased *EC* relative to the holo replicates (Fig 10B and S17 Fig). As these regions are around the active site, their decreased *EC* in the PfIleRS-SANC systems may affect enzymatic function, especially for some of the RF-binding compounds which, as earlier discussed, affected the fluctuation of key residues of the KMSKR and HYGH motifs and perhaps consequentially, the AMP binding affinity. In contrast, the editing domain has very consistent holo replicate values and considerably increased *EC* in pocket 1 PfIleRS-SANC systems **SANC140**, **SANC968**, and **SANC1126** (Fig 10B and S17 Fig). Therefore, these RF-binding (pocket 1) ligands seem to induce increased residue communication in the editing domain, especially **SANC140**. For the sake of completeness, all systems' top/bottom 3% delta *EC* residues are mapped to structure as spheres in S18 Fig. Moving on, most PfIleRS-SANC systems have increased *EC* of regions I797-L821 (includes α26), G827-V832 (ABD; α27), and F955-Y962 (ABD). These regions are mostly located in the *EC* group on the right (Fig 10A). Thus, in most PfIleRS-SANC systems, the high *EC* active site group on the left exhibited decreased *EC* while the high *EC* active site group on the right exhibited

increased *EC* (S17 Fig). There may be a strong interplay between these two high *EC* regions, reaching across the active site. Lastly, the PfIleRS-SANC systems in pockets 3–6 show increased *EC* in residues Y139-R153 (α4; part of HYGH motif) and residues located within the bottom of the active site, namely residues I740-T747, and L770-L777. All of these large *EC* increases in the active site of ligand-bound systems may be a result of the SANC ligands' disruptive effects on the active site and constituent substrates as established in earlier RMSD, RMSF, and AMP energy calculations.

*Closeness centrality* (*CC*) measures the inverse of the mean distance of node *i* to all other nodes, therefore the nodes with the highest *CC* values are positioned at the geometric "centre" of the graph [118]. In S19 Fig, residues with the largest *CC* values are found within and around the active site, approximately in the geometric centre of the enzyme as per the definition of *CC*. Delta *CC* therefore indicates the change of location of the residues relative to the geometric centre. The delta *CC* heatmap (Fig 11) shows that many regions of most SANC systems have large *CC* decreases, indicating that these regions are spending most of the trajectory further away from the geometric centre relative to the holo average. Further, the colour bar shows that the largest *CC* increase overall is 0.0095, whereas the largest *CC* decrease is much larger with 0.0209. Many SANC systems, particularly those binding at the RF pockets, showed decreased *CC* of active site residues (including the HYGH and KMSKR motifs), zinc finger residues, and potential allosteric pocket 1 residues W105-K119 (α-helix 2→α-helix 3).

Fig 12 shows the delta *CC* of the RF-binding **SANC456** which is the representative structure shown as this system saw the greatest *CC* changes. The structures of the active site, zinc fingers, and start of the editing domain are not aligned with the three holo replicates (Fig 12A). The decreased *CC* (blue) is an indication that these residues are spending most of the trajectory further away from the geometric centre relative to the holo average, which corresponds to the structure clustering. The large structural changes of the active site could affect binding and catalysis, while structural changes of the two zinc finger motifs could indicate interference with interdomain communication. Also, the large structural changes of the two anti-parallel beta-sheets in the editing domain (Y281→I294 and M567→Y576) could affect the linkage to the Rossmann fold catalytic domain. The stable binding of **SANC456** in potential allosteric pocket 1 may cause the structural and positional changes of α-helix 2 and α-helix 3 (Fig 12B), which may lead to the large structural changes/delta *CC* seen elsewhere (Fig 12C and S20 Fig). This is likely linked to: 1) the pathway of increased *BC* residues going from

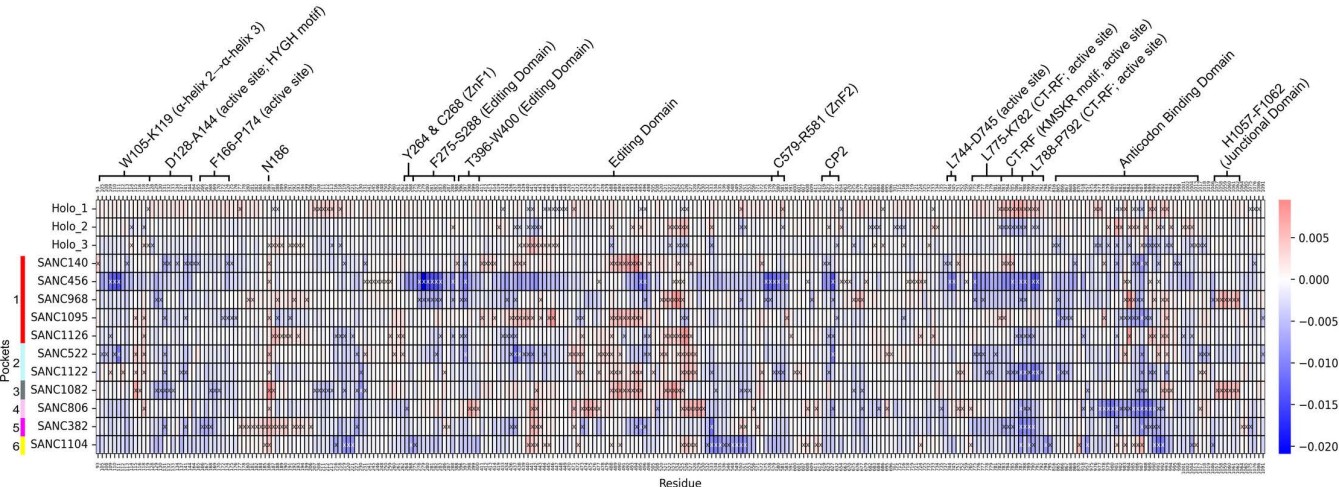

**Fig 11. Delta *CC* heatmap.** The blue colour indicates a decrease from the holo-averaged protein, while the red colour indicates an increase. Residues per system with the largest change in *CC* (top/bottom 3% delta: 54 residues) relative to holo are shown with an "x". Residues in the heatmap not shown with an "x" are not in the top/bottom 3% in their respective system; these are included for comparison purposes. Many SANC systems show decreased *CC* of zinc finger and active site residues, including the HYGH and KMSKR motifs.

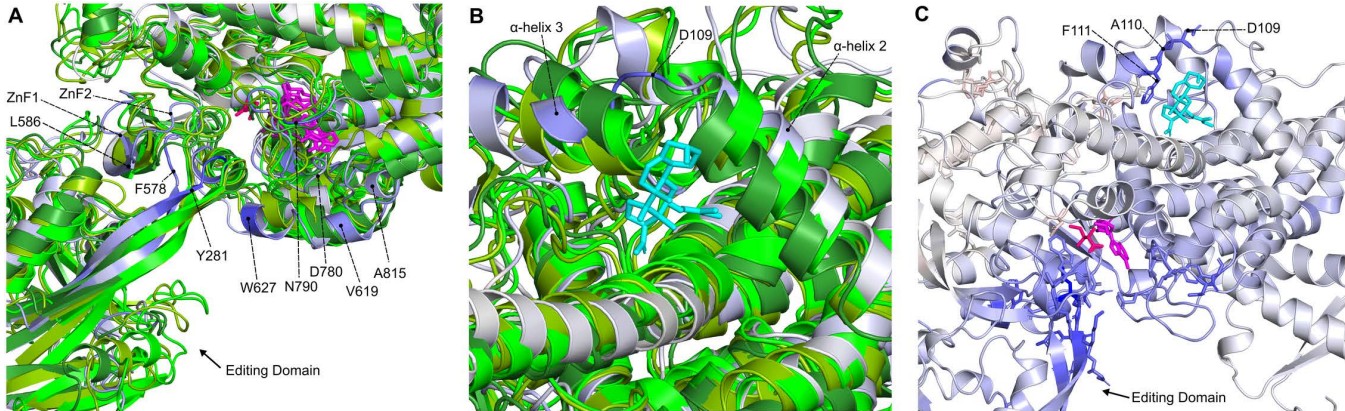

**Fig 12. PfIleRS-SANC456 delta *CC*.** Superimposed structures of holo systems (green) and PfIleRS-SANC456 (delta colours) showing delta *CC* values. AMP and SANC ligands are coloured magenta and cyan, respectively. Holo and PfIleRS-SANC456 Ile ligands are coloured purple and pink, respectively. A) The structure of the active site, zinc fingers, and start of the editing domain are not aligned with the three holo replicates. The decreased *CC* (blue) is an indication that these regions are spending most of the trajectory further away from the geometric centre relative to the holo average, which corresponds to the structure clustering. B&C) The stable binding of SANC456 in potential allosteric pocket 1 may cause the structural and positional changes of α-helix 2 and α-helix 3 which may lead to the large structural changes/delta *CC* seen elsewhere.

potential allosteric pocket 1 into the active site, 2) the increased distance between the Ile and AMP ligands of nearly 15 Å, 3) decreased AMP binding affinity, and 4) increased RMSF of many key active site residues in the KMSKR and HYGH motifs.

Although not as extreme as **SANC456**, according to the heatmap (Fig 11) most SANC systems are exhibiting structural changes/delta *CC* not seen in the holo replicates, most importantly being the decreased *CC* of zinc finger residues and active site residues (including the HYGH and KMSKR motifs).

## 4. Conclusions

Increasing *Pf* resistance to first-line antimalarial therapies presents an urgent need to explore new Plasmodium drug targets and develop potent solutions against multidrug-resistant strains. aaRSs are present in all developmental stages of the parasites and therefore have been reported as ideal drug targets for combating drug resistance. In this study, we focused on IleRS as a target PfaaRS of interest and modelled high-quality structures of both the plasmodial protein and human homolog. We then identified potential allosteric and druggable pockets in PfIleRS that could be bound by small molecule compounds. Thereafter, we used blind-docking to screen a total of 1019 SANCDB compounds against PfIleRS and cross-screened against the human protein to assess the selectivity potential of our hits against the parasite protein. Eleven SANCDB were identified as potential hits with higher binding affinity towards PfIleRS than HsIleRS. MD simulations of protein-compound complexes were performed and post MD analyses identified possible allosteric effects and mechanisms. Our findings can be summarised as follows: **(1)** Among the 11 PfIleRS-selective SANCDB compounds; **SANC140**, **SANC456**, **SANC968**, **SANC1095**, and **SANC1126** preferentially targeted a pocket at the N-terminal Rossmann fold; **SANC522** and **SANC1122** were bound at C-terminal Rossmann fold; **SANC806** docked to the zinc finger hinge; **SANC382** was bound near the connective peptide 1; **SANC1082** and **SANC1104** docked to separate pockets at the anticodon binding domain. **(2)** Each of these target sites have allosteric potentials and are druggable as predicted and validated by multiple pocket prediction tools. **(3)** Each of the compounds satisfied ADMET and passed the PAINS, suggesting that these compounds are potentially suitable for further optimisation and pre-clinical evaluation. SANC382, SANC522, SANC806, SANC968, SANC1095, SANC1126 also showed potential drug lead-like properties. **(4)** PfIleRS presented different RMSD profiles in the presence of some of the SANCDB compounds when compared to the holo systems. The most drastic

conformational changes were observed in the **PfIleRS-SANC456** complex. **SANC456** was previously identified as a potential allosteric modulator of PfProRS protein [11]. This could potentiate **SANC456** as a possible pan-aaRS inhibitor. **(5)** The compounds, even though binding distantly from the catalytic site, impacted the binding affinity of the key nucleotide substrate, AMP, particularly **SANC456**, which also substantially increased the distance between the catalytic AMP and Ile substrates. This could suggest a mechanism by which these ligands can possibly impact the aminoacylation activities of the protein. **(6)** The binding activities of the SANCDB hit compounds, according to our DRN calculations, induced notable *BC*, *EC,* and *CC* changes that interfered with the characteristic communication path between critical protein domains (editing, catalytic Rossmann fold, anticodon binding, and junctional domains) and affected key regions around the active sites that could be disruptive to the optimal conformation and dynamics favourable for enzymatic (aminoacylation) activities. **(7)** Overall, the introduction of the SANC ligands to the PfIleRS enzyme induced changes in residue communication pathways, which may have resulted in the active site structural alterations and increased active site RMSD relative to the holo enzyme. These could also account for the increased distance between the catalytic AMP and Ile substrates seen in some SANC systems, and the decreased AMP binding affinity seen in all SANC systems. These, altogether, hypothesise the potential allosteric effects of the identified hits on the structure and function of PfIleRS.

The next phase of our study aims to facilitate *in vitro* enzymatic inhibition assays to validate and confirm allosteric activity of these compounds as well as their analogues.

## Supporting information

**S1 File.   S1 Fig.** PDB template structures validation report and PfIleRS and HsIleRS verify3D and ProSA validation reports. (A) 7D5C template resolution of 1.9 Å and Rfree of 0.20. (B) 6LDK Resolution of 2.9 Å and and Rfree of 0.24. (C) PfIleRS and HsIleRS scatter plots showing the percentage of residues that scored >= 0.1 in the 3D/1D profile. (D) The position of the modelled Plasmodium and human structures in references to experimentally determined structures (X-ray and Nuclear Magnetic Resonance (NMR). **S2 Fig.** Conservation scores of all predicted pockets generated from multiple sequence alignment of PfIleRS with other species of plasmodium: (*Plasmodium malariae*, *Plasmodium knowlesi*) and mammalian counterparts (*Homo sapiens* and *Mus musculus*). Normalised conservation scores are indicated for each residue in the alignment. **S3 Fig.** Ligplot analysis of modelled and redocked AMP complex interactions. (A) PfIleRS-AMP modelled and redocked residue interactions. (B) HsIleRS-AMP modelled and redocked residue interactions. The modelled AMP (grey) and redocked AMP (blue); while identical interacting residues are shown in red ellipses. **S4 Fig.** Protein-ligand interactions of selected potential SANCDB hit compounds against PfIleRS protein. All possible interactions formed between the PfIleRS receptor and the SANCDB compound hits are shown with their respective binding modes. **S5 Fig.** Binding modes and protein-ligand interactions of SANCDB potential hits in HsIleRS protein. **S6 Fig.** Global trajectory analysis of PfIleRS, PfIleRS orthosteric site and HsIleRS holo and ligand bound complexes. (A) Backbone-RMSD line plots of holo states and PfIleRS-SANC complexes. (B) Orthosteric pocket backbone-RMSD line plots for holo and PfIleRS-SANC systems. (C) Back-bone RMSD line plots for HsIleRS holo and ligand complexes. **S7 Fig.** PfIleRS-SANDB complexes superimposed trajectory snap shots of the 3 most sampled conformations. Dotted circles indicate the regions of PfIleRS structure where significant RMSD deviations were observed. **S8 Fig.** PfIleRS structural conformation cluster 1 of each system from the structure clustering. Systems SANC140, SANC522, SANC806, and SANC1095 have a secondary structure change (black dashed circles) of the two anti-parallel beta-sheets in the editing domain (Y281→I294 and M567→Y576) that link the domain to the rest of the enzyme. Importantly, these anti-parallel beta-sheets have consistent secondary structure in the holo systems. **S9 Fig.** Superimposed HsIleRS-SANC complexes structures of the 3 most sampled conformations across the trajectory. The dotted circles indicated regions in the HsIleRS structure that reported significant RMSD deviations. **S10 Fig.** HsIleRS ligand-RMSD line plots for identified potential SANCDB hits over 100 ns MD simulations. **S11 Fig.** HsIleRS holo states and HsIleRS-SANC RMSF line plots. The atomic-residue fluctuations in

HsIleRS structure are divided according to the domains. (A) The N-terminal Rossmann fold, (B) editing domain, (C) the C-terminal Rossmann fold, and (D) the anticodon binding domain. **S12 Fig.** Delta RMSF values mapped to structure. Top/bottom 3% delta RMSF residues per system are shown as spheres (54 residues). The blue colour indicates a decrease from the holo-averaged protein, while the red colour indicates an increase. AMP, Ile and SANC ligands are coloured magenta, green and cyan, respectively. **S13 Fig.** Center of mass (COM) distance between the Ile and AMP ligands over 100 ns trajectory in holo and PfIleRS-SANC complex systems. **S14 Fig.** Delta BC. The blue colour indicates a decrease from the holo-averaged protein, while the red colour indicates an increase. A) Structure of SANC456, representing pocket 1 SANC systems plus SANC522 (pocket 2). Spheres show residues mentioned in the main text, with the remaining top/bottom 3% delta BC residues shown as sticks. A decreased BC pathway between the active site and editing domain can be seen, indicating decreased communication between these regions. B) Structure of SANC806. Spheres show top/bottom 3% delta BC residues. Increased BC residues form a pathway starting near the SANC806 ligand pocket in the Editing Domain, moving through the active site and towards the Junctional Domain through Anticodon Binding Domain residues. This may affect the flow of information within the enzyme. **S15 Fig.** Delta BC values mapped to structure. Top/bottom 3% delta BC residues per system are shown as spheres (54 residues). The blue colour indicates a decrease from the holo-averaged protein, while the red colour indicates an increase. AMP, Ile and SANC ligands are coloured magenta, green and cyan, respectively. **S16 Fig.** PfIleRS-SANC systems with top/bottom 3% delta BC residues forming potential allosteric paths from the SANC ligand binding site to the active site. Top/bottom 3% delta BC residues shown as spheres. The blue colour indicates a decrease from the holo-averaged protein, while the red colour indicates an increase. AMP, Ile and SANC ligands are coloured magenta, green and cyan, respectively. **S17 Fig.** Delta EC. Representative structures of SANC968 and SANC1082 used to show regions of large EC changes (black blocks). Top/bottom 3% delta EC residues shown as spheres. The blue colour indicates a decrease from the holo-averaged protein, while the red colour indicates an increase. AMP, Ile and SANC ligands are coloured magenta, green and cyan, respectively. In most PfIleRS-SANC systems, the high EC active site group on the left exhibited decreased EC while the high EC active site group on the right exhibited increased EC. **S18 Fig.** Delta EC values mapped to structure. Top/bottom 3% delta EC residues per system shown as spheres (54 residues). The blue colour indicates a decrease from the holo-averaged protein, while the red colour indicates an increase. AMP, Ile and SANC ligands are coloured magenta, green and cyan, respectively. **S19 Fig.** Holo average top 5% CC values mapped to structure and shown in a bar plot. AMP and Ile ligands are coloured magenta and green, respectively. Residues with the largest CC values are found within and around the active site, approximately in the geometric centre of the enzyme as per the definition of CC. The holo CC values mapped to structure were normalised to start at zero for visualisation purposes, due to there being a very small difference between the highest and lowest un-normalised CC values. **S20 Fig.** Delta CC values mapped to structure. Top/bottom 3% delta CC residues per system are shown as spheres (54 residues). The blue colour indicates a decrease from the holo-averaged protein, while the red colour indicates an increase. AMP, Ile and SANC ligands are coloured magenta, green and cyan, respectively. **S1 Table.** Parameters used in cross-docking validation and blind docking of PfIleRS and HsIleRS proteins. **S2 Table.** Orthosteric and allosteric consensus residues of all the predicted pockets in PfIleRS using SiteMap, DogSiteScorer, FTMap and PASSer prediction algorithms. **S3 Table.** Orthosteric and allosteric consensus residues of all the predicted pockets in HsIleRS using SiteMap, DogSiteScorer, FTMap and PASSer prediction algorithms. **S4 Table.** Sequence, structural and quantitative differences between PfIleRS and HsIleRS potential allosteric pockets. **S5 Table.** Active inhibitors collated from literature for PfIleRS protein used in docking protocol validation. **S6 Table.** Absorption, Distribution, Metabolism, Excretion and Toxicity (ADMET) properties for SANCDB natural products selected as potential hits against PfIleRS. **S7 Table.** RMSD mean values of Holo and PfIleRS-SANC complexes over 100 ns MD simulations. **S8 Table.** HSIleRS-SANC complexes mean RMSD values over 100 ns MD simulations. **S9 Table.** PfIleRS-SANC conformations frequency for each structure cluster. **S10 Table.** HsIleRS-SANC frequencies of most sampled conformations in each structure cluster. **S11 Table.** Holo and PfIleRS-SANC complexes binding free energy terms for AMP ligands as determined by molecular

mechanics Generalised-Born surface area (MM-GBSA) analysis. **S12 Table.** Binding free energy terms estimations of SANCDB compounds in complex with PfIleRS protein.
(PDF)

## Acknowledgements

The authors thank the Centre for High Performance Computing (CHPC), Cape Town, South Africa for the computational resources. The authors further thank Prof. Kevin Lobb and Rabelani Ramahala for their contributions in writing and troubleshooting molecular docking scripts. Curtis Chepsiror thanks Queen Elizabeth Commonwealth Scholarships (QECS) and Association of Commonwealth Universities (ACU) for provision of scholarship bursary to study at Rhodes University.

## Author contributions

**Conceptualization:** Özlem Tastan Bishop.

**Data curation:** Curtis Chepsiror.

**Formal analysis:** Curtis Chepsiror, Wayde Veldman, Fisayo Olotu, Özlem Tastan Bishop.

**Funding acquisition:** Özlem Tastan Bishop.

**Investigation:** Curtis Chepsiror, Özlem Tastan Bishop.

**Methodology:** Curtis Chepsiror, Wayde Veldman, Fisayo Olotu, Özlem Tastan Bishop.

**Project administration:** Özlem Tastan Bishop.

**Resources:** Özlem Tastan Bishop.

**Supervision:** Fisayo Olotu, Özlem Tastan Bishop.

**Validation:** Curtis Chepsiror, Wayde Veldman, Özlem Tastan Bishop.

**Visualization:** Curtis Chepsiror, Wayde Veldman, Özlem Tastan Bishop.

**Writing – original draft:** Curtis Chepsiror, Wayde Veldman, Özlem Tastan Bishop.

**Writing – review & editing:** Curtis Chepsiror, Wayde Veldman, Fisayo Olotu, Özlem Tastan Bishop.

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
