## [Decision Letter · Decision Letter 0]

2 Jan 2025

PONE-D-24-49827Allosteric Modulation of Plasmodium falciparum Isoleucyl tRNA Synthetase by South African Natural CompoundsPLOS ONE

Dear Dr. Tastan Bishop,

Thank you for submitting your manuscript to PLOS ONE. After careful consideration, we feel that it has merit but does not fully meet PLOS ONE’s publication criteria as it currently stands. Therefore, we invite you to submit a revised version of the manuscript that addresses the points raised during the review process.

We look forward to receiving your revised manuscript.

Kind regards,

Opeyemi Iwaloye

Academic Editor

PLOS ONE

Journal Requirements:

Reviewers' comments:

Reviewer's Responses to Questions

**Comments to the Author**

1. Is the manuscript technically sound, and do the data support the conclusions?

Reviewer #1: Yes

Reviewer #2: Yes

Reviewer #3: Yes

Reviewer #4: Yes

2. Has the statistical analysis been performed appropriately and rigorously? 

Reviewer #1: Yes

Reviewer #2: Yes

Reviewer #3: Yes

Reviewer #4: N/A

3. Have the authors made all data underlying the findings in their manuscript fully available?

Reviewer #1: Yes

Reviewer #2: No

Reviewer #3: Yes

Reviewer #4: Yes

4. Is the manuscript presented in an intelligible fashion and written in standard English?

Reviewer #1: Yes

Reviewer #2: Yes

Reviewer #3: Yes

Reviewer #4: Yes

5. Review Comments to the Author

Reviewer #1: The manuscript titled "Allosteric Modulation of Plasmodium falciparum Isoleucyl tRNA Synthetase by South African Natural Compounds" explores the use of in-silico methods to analyze the activities of SANC against multi-drug resistant malaria parasite.

The authors reported that their "MD simulation studies revealed that the compounds commonly induced changes in the global conformation and dynamics of PflleRS, particularly SANC456, indicative of their possible allosteric modulatory effects. The homology modeling, molecular docking and molecular dynamics simulations were conducted using standardized protocols. The post analytic statistical analysis were conducted appropriately and rigorously.

The authors should add all figures (Figs. 1 - 9) in the appropriate sections.

Reviewer #2: The manuscript investigates allosteric inhibitors targeting Plasmodium falciparum Isoleucyl tRNA synthetase (PfIleRS) using computational methods. It employs homology modeling, molecular docking, molecular dynamics simulations, and residue network analysis to screen natural compounds from South Africa for antimalarial activity. The study identifies 11 promising compounds, including SANC456, SANC1095, and SANC140, which show selective binding and potential as modulators.

While the research addresses critical issues like drug resistance in malaria and uses innovative approaches, it could benefit from improvements in clarity, depth, and practical application.

1. The study lacks in vitro or in vivo validation, which is essential to confirm the computational predictions. The authors should discuss plans or preliminary steps toward experimental validation, such as enzymatic inhibition assays or surface plasmon resonance studies. In the "Discussion" section (lines 157–159), the authors state: "We believe that compounds reported in this study can serve as reliable starting points... I believe it would be much better to propose experiments to investigate this such as enzymatic inhibition assays to test the ability of SANC456, SANC1095, and SANC140 to modulate PfIleRS activity.

2. Molecular dynamics trajectories are only available upon request, potentially limiting reproducibility. It will be better to deposit all molecular dynamics trajectories and associated files in a public repository like Zenodo or Dryad and include accessible links in the manuscript.

3. The methodology section is longwinded and includes redundant details, particularly on docking parameters. Streamline the methods by summarizing repetitive details and moving extended technical parameters to supplementary materials. In Section 2.3.2 (lines 214–229), the authors describe "Prepare_receptor4.py and prepare_ligand4.py scripts... The PfIleRS box size was 110.60 Å x 87.12 Å...". Why not streamline this by following an approach such as this “"Docking parameters were optimized using AutoDock Vina with grid box sizes defined based on receptor dimensions (see Supplementary Table S1)."

4. The differences between PfIleRS and human IleRS allosteric pockets are discussed but lack quantitative details to highlight specificity. Include a comparative table of sequence/structural differences between PfIleRS and human IleRS at the allosteric level.

5. Some findings overlap with the authors' prior publication on homology modeling and pocket detection. Clearly distinguish the novel contributions of this study and minimize redundancy.

6. In the Introduction (lines 368–370):"In our recent article, we reported homology modeling and identification of potential allosteric pockets..."

How about discussing the differences such as "Building on our previous work on homology modeling, this study expands the pipeline by integrating dynamic residue network analysis to identify novel allosteric modulators."

7. Figures illustrating docking interactions and dynamic residue networks are overly complex. Simplify visuals by focusing on key findings. For example, highlight only the top 3 compounds and their binding modes in Figure 5.

8. The translational relevance of findings is understated. Expand on how the identified compounds could be optimized for clinical development, including potential ADMET properties.

Reviewer #3: The manuscript is well-written and addresses a significant concern in the health care system by targeting Plasmodium falciparum aminoacyl tRNA synthetase, a viable strategy to overcome malaria parasite multi-drug resistance.

The manuscript should be accepted for publication. However, I suggest addressing these concerns to further enhance its impact.

1. The figure resolution is unclear, making the diagram quite difficult to read.

2. Kindly include future directions in the manuscript.

Reviewer #4: I found this manuscript to be a timely and innovative contribution to antimalarial drug discovery, particularly in its approach to targeting Plasmodium falciparum Isoleucyl-tRNA Synthetase (PfIleRS) with allosteric inhibitors. I appreciate the methodological rigor demonstrated in the study, which integrates molecular docking, MD simulations, binding free energy calculations, and dynamic residue network analysis. The findings offer valuable mechanistic insights into how allosteric modulation disrupts catalytic function, with detailed residue-level analyses highlighting key players in enzymatic activity and communication pathways. I find the focus on species-specific inhibitors particularly compelling, as it underscores the potential to minimize off-target effects. Additionally, the novelty of targeting allosteric pockets enhances the therapeutic relevance of this work, making it a significant contribution to the field. In addition, the study does not require any statistical analysis.

6. PLOS authors have the option to publish the peer review history of their article (what does this mean? ). If published, this will include your full peer review and any attached files.

**Do you want your identity to be public for this peer review?** For information about this choice, including consent withdrawal, please see our Privacy Policy .

Reviewer #1: No

Reviewer #2: No

Reviewer #3: No

Reviewer #4: No

---

## [Author Response · Author response to Decision Letter 0]

6 Feb 2025

Dear Editor,

We thank the reviewers for their constructive comments. We have revised the manuscript accordingly and believe that their suggestions have significantly improved its quality. We hope that our revised version meets the expectations of both the reviewers and the journal.

A summary of our revisions, along with individual responses to each comment, can be found on the following pages.

Reviewer #1:

The manuscript titled "Allosteric Modulation of Plasmodium falciparum Isoleucyl tRNA Synthetase by South African Natural Compounds" explores the use of in-silico methods to analyze the activities of SANC against multi-drug resistant malaria parasite.

The authors reported that their "MD simulation studies revealed that the compounds commonly induced changes in the global conformation and dynamics of PflleRS, particularly SANC456, indicative of their possible allosteric modulatory effects. The homology modeling, molecular docking and molecular dynamics simulations were conducted using standardized protocols. The post analytic statistical analysis were conducted appropriately and rigorously.

1. The authors should add all figures (Figs. 1 - 9) in the appropriate sections.

Response: We thank the reviewer for the kind comments about our manuscript. The PLOS ONE journal recommends the figures file to be submitted to the journal as a separate file; thereafter, all the figures will be added to the appropriate sections by the journal production editors once the manuscript is accepted.

Reviewer #2:

The manuscript investigates allosteric inhibitors targeting Plasmodium falciparum Isoleucyl tRNA synthetase (PfIleRS) using computational methods. It employs homology modeling, molecular docking, molecular dynamics simulations, and residue network analysis to screen natural compounds from South Africa for antimalarial activity. The study identifies 11 promising compounds, including SANC456, SANC1095, and SANC140, which show selective binding and potential as modulators.

While the research addresses critical issues like drug resistance in malaria and uses innovative approaches, it could benefit from improvements in clarity, depth, and practical application.

1. The study lacks in vitro or in vivo validation, which is essential to confirm the computational predictions. The authors should discuss plans or preliminary steps toward experimental validation, such as enzymatic inhibition assays or surface plasmon resonance studies. In the "Discussion" section (lines 157–159), the authors state: "We believe that compounds reported in this study can serve as reliable starting points... I believe it would be much better to propose experiments to investigate this such as enzymatic inhibition assays to test the ability of SANC456, SANC1095, and SANC140 to modulate PfIleRS activity.

Response: We agree that in vitro or in vivo experimental validation will significantly improve the impact of our findings. We changed the sentence in the abstract as you suggested to emphasize the importance of experimental data. It now reads:

“We believe that the compounds identified in this study as potential allosteric inhibitors have strong translational potential and warrant further investigation through in vitro and in vivo experiments. Overall, they hold promise as starting points for the development of new and effective antimalarial therapies, particularly against multidrug-resistant Plasmodium parasites.”

We also included a concluding sentence in the Conclusion.

2. Molecular dynamics trajectories are only available upon request, potentially limiting reproducibility. It will be better to deposit all molecular dynamics trajectories and associated files in a public repository like Zenodo or Dryad and include accessible links in the manuscript.

Response: We appreciate the reviewer for this suggestion as it significantly enhances the reproducibility of our manuscript.

We have uploaded all the molecular dynamics trajectories for all the systems that were analysed in the study with the intermittent csv files with raw data, scripts, modules, libraries and protocols used in analysis with a permanent digital object identifier: https://doi.org/10.5061/dryad.d7wm37qb6. The data will be publicly available once the manuscript is published, but here is the reviewer URL link to access the data currently: http://datadryad.org/stash/share/VXMBKruJLgVbVo4TpK7123hxoepH4EQbJuUxy6Yi16o.

3. The methodology section is longwinded and includes redundant details, particularly on docking parameters. Streamline the methods by summarizing repetitive details and moving extended technical parameters to supplementary materials. In Section 2.3.2 (lines 214–229), the authors describe “Prepare_receptor4.py and prepare_ligand4.py scripts… The PfIleRS box size was 110.60 Å x 87.12 Å…”. Why not streamline this by following an approach such as this “”Docking parameters were optimized using AutoDock Vina with grid box sizes defined based on receptor dimensions (see Supplementary Table S1).”

Response: We thank the reviewer for this suggestion. Section 2.3.2 paragraph 1 has been streamlined to remove repetitive details as shown in our revised manuscript. The technical details have been moved to the supplementary data S6 Table and indicated in the first sentence of the paragraph as suggested by the reviewer.

4. The differences between PfIleRS and human IleRS allosteric pockets are discussed but lack quantitative details to highlight specificity. Include a comparative table of sequence/structural differences between PfIleRS and human IleRS at the allosteric level.

Response: We thank the reviewer for this important suggestion. We have added a table that illustrates sequence/structural differences between PfIleRS and HsIleRS predicted allosteric pockets (S3 Table) and discussed the details in Section 3.1 paragraph 6. The table shows conserved residues in Plasmodium and human IleRS, structural difference in size, number of hydrophobic residues, and quantitative differences in complementary residues in each of the IleRS proteins.

Additional text is as follows:

“Multiple sequence alignment of PfIleRS and HsIleRS further revealed differences in the amino acid compositions of these pockets (S2 Fig). Conserved allosteric residues in PfIleRS and HsIleRS exhibited structural and sequence-based quantitative differences suggesting that the predicted pockets possess unique features in Pf and human IleRS (S3 Table). Predicted pockets in PfIleRS and HsIleRS showed variations in physio-chemical properties in the conserved residues in terms of polarity, size, charge and hydrophobicity. S3 Table indicates that PfIleRS has high hydrophobicity compared to HsIleRS, while the pocket size in PfIleRS was structurally smaller than in HsIleRS. These dissimilarities favour the formation of bond interactions that are unique and specific to PfIleRS. The potential to specifically target PfIleRS with mitomycin via its editing domain pocket over HsIleRS has been previously exploited [7].”

5. Some findings overlap with the authors' prior publication on homology modeling and pocket detection. Clearly distinguish the novel contributions of this study and minimize redundancy.

Response: Our allosteric pocket detection results have been revised in Section 3.1 paragraph 2 in lines (381-385) to provide a clear distinction from our previous article. More details have been provided on the different approaches used to maximise the number of pockets detected.

It now reads:

“ …. Furthermore, in addition to employing multiple prediction algorithms to detect potential pockets, we utilised residue conservation analysis to identify potent sites suitable for ligand binding (S2 Fig). Pocket residues are identified by using an in-house script and predicted pockets were used to guide the compound’s selection process to identify potential hits. Information on the residue composition of each identified pocket is provided in S1 and S2 Tables, obtained by defining residues within 5 Å of the pocket centroid based on consensus from multiple prediction programs.”

6. In the Introduction (lines 368–370):"In our recent article, we reported homology modeling and identification of potential allosteric pockets..." How about discussing the differences such as "Building on our previous work on homology modeling, this study expands the pipeline by integrating dynamic residue network analysis to identify novel allosteric modulators."

Response: We thank the reviewer for this comment. We have revised the opening sentence in Section 3.1 paragraph 1 from "In our recent article, we reported homology modeling and identification of potential allosteric pockets..." to " Building on our previous work, this study expands the approach by integrating DRN analysis to identify the effect of novel potential allosteric modulators”.

7. Figures illustrating docking interactions and dynamic residue networks are overly complex. Simplify visuals by focusing on key findings. For example, highlight only the top 3 compounds and their binding modes in Figure 5.

Response: Figure 5 has been revised to reflect only the top compounds that were bound to potential allosteric site 1 which had the most impact. The original Figure 5 moved to the Supplementary Document.

8. The translational relevance of findings is understated. Expand on how the identified compounds could be optimized for clinical development, including potential ADMET properties.

Response: We thank the reviewer for this constructive suggestion. ADMET properties have been included in the methodology Section 2.3.2 paragraph 3 and discussion Section 3.3 paragraph 1 to show the potential of identified compounds to induce in vitro allosteric activity and early phase clinical development.

Further translational relevance of the findings has been stated in the conclusion section. It reads:

“…. (3) Each of the compounds satisfied ADMET and passed the PAINS, suggesting that these compounds are potentially suitable for further optimisation and pre-clinical evaluation. SANC382, SANC522, SANC806, SANC968, SANC1095, SANC1126 also showed potential drug lead-like properties….”

Reviewer #3:

The manuscript is well-written and addresses a significant concern in the health care system by targeting Plasmodium falciparum aminoacyl tRNA synthetase, a viable strategy to overcome malaria parasite multi-drug resistance.

The manuscript should be accepted for publication. However, I suggest addressing these concerns to further enhance its impact.

1. The figure resolution is unclear, making the diagram quite difficult to read.

Our response: We thank the reviewer for the constructive comments to improve our manuscript. All our main figures have been revised and resolution improved. Further they have been uploaded to Preflight Analysis and Conversion Engine (PACE) to ensure they meet PLOS requirements.

2. Kindly include future directions in the manuscript.

Our response: We have included the future directions in the last section of Conclusions. It reads:

“The next phase of our study aims to facilitate in vitro enzymatic inhibition assays to validate and confirm allosteric activity of these compounds as well as their analogues.”

Reviewer #4:

I found this manuscript to be a timely and innovative contribution to antimalarial drug discovery, particularly in its approach to targeting Plasmodium falciparum Isoleucyl-tRNA Synthetase (PfIleRS) with allosteric inhibitors. I appreciate the methodological rigor demonstrated in the study, which integrates molecular docking, MD simulations, binding free energy calculations, and dynamic residue network analysis. The findings offer valuable mechanistic insights into how allosteric modulation disrupts catalytic function, with detailed residue-level analyses highlighting key players in enzymatic activity and communication pathways. I find the focus on species-specific inhibitors particularly compelling, as it underscores the potential to minimize off-target effects. Additionally, the novelty of targeting allosteric pockets enhances the therapeutic relevance of this work, making it a significant contribution to the field. In addition, the study does not require any statistical analysis.

Response: Authors thank the reviewer for the kind comments made concerning our manuscript. Indeed, non-competitive inhibition of Plasmodium drug targets with species-specific inhibitors is the most viable way to combat multidrug resistance encountered by the current antimalarials.

---

## [Editor Report · Decision Letter 1]

7 Mar 2025

Allosteric Modulation of Plasmodium falciparum Isoleucyl tRNA Synthetase by South African Natural Compounds

PONE-D-24-49827R1

Dear Dr. Chepsiror,

We’re pleased to inform you that your manuscript has been judged scientifically suitable for publication and will be formally accepted for publication once it meets all outstanding technical requirements.

Kind regards,

Opeyemi Iwaloye

Academic Editor

PLOS ONE
---

## [Editor Report · Acceptance letter]

PONE-D-24-49827R1

PLOS ONE

Dear Dr. Tastan Bishop,

I'm pleased to inform you that your manuscript has been deemed suitable for publication in PLOS ONE. Congratulations! Your manuscript is now being handed over to our production team.

Kind regards,

on behalf of

Dr. Opeyemi Iwaloye

Academic Editor

PLOS ONE